# Chromatin targeting of the RNF12/RLIM E3 ubiquitin ligase controls transcriptional responses

Carmen Espejo-Serrano[1], Catriona Aitken[1], Beatrice F Tan[2], Danielle G May[3], Rachel J Chrisopulos[3], Kyle J Roux[3,4], Jeroen AA Demmers[5], Samuel G Mackintosh[6], Joost Gribnau[2], Francisco Bustos[4,7], Cristina Gontan[2], Greg M Findlay[1]

**Protein ubiquitylation regulates key biological processes including transcription. This is exemplified by the E3 ubiquitin ligase RNF12/RLIM, which controls developmental gene expression by ubiquitylating the REX1 transcription factor and is mutated in an X-linked intellectual disability disorder. However, the precise mechanisms by which ubiquitylation drives specific transcriptional responses are not known. Here, we show that RNF12 is recruited to specific genomic locations via a consensus sequence motif, which enables co-localisation with REX1 substrate at gene promoters. Surprisingly, RNF12 chromatin recruitment is achieved via a non-catalytic basic region and comprises a previously unappreciated N-terminal autoinhibitory mechanism. Furthermore, RNF12 chromatin targeting is critical for REX1 ubiquitylation and downstream RNF12-dependent gene regulation. Our results demonstrate a key role for chromatin in regulation of the RNF12-REX1 axis and provide insight into mechanisms by which protein ubiquitylation enables programming of gene expression.**

## Introduction

Protein ubiquitylation is a critical post-translational modification that controls all aspects of biology (Kulathu & Komander, 2012; Oh et al, 2018). As a result, E3 ubiquitin ligases, which select substrates for ubiquitylation, serve as regulatory gatekeepers of myriad biological processes, including biologically critical functions such as protein homeostasis and quality control, cell cycle, and the DNA damage response (Kulathu & Komander, 2012; Oh et al, 2018). Ubiquitylation also orchestrates signalling events, for example, in immune cell signalling (Bhoj & Chen, 2009; Popovic et al, 2014). Therefore, dysregulation of E3 ubiquitin ligases has been implicated in many human diseases, such as cancer, disorders of the immune system, and developmental disorders (Ciechanover & Brundin, 2003; Popovic et al, 2014; Rape, 2018).

A key function of protein ubiquitylation is in control of gene expression and cell identity, decision-making processes that frequently go awry in disease. This is exemplified by the E3 ubiquitin ligase RNF12/RLIM, which controls developmental gene expression (Zhang et al, 2012; Bustos et al, 2020; Segarra-Fas et al, 2022), mammary gland development and function (Jiao et al, 2012), and X-chromosome inactivation (Jonkers et al, 2009; Shin et al, 2010; Barakat et al, 2011; Gontan et al, 2012, 2018). RNF12 is mutated in the X-linked intellectual disability disorder Tonne-Kalscheuer syndrome (TOKAS) (Tønne et al, 2015; Hu et al, 2016; Frints et al, 2019; Bustos et al, 2021). RNF12 variants identified in TOKAS patients disrupt E3 ubiquitin ligase activity (Bustos et al, 2018; Frints et al, 2019), suggesting that an RNF12-dependent ubiquitin signalling pathway goes awry to cause intellectual disability in these individuals.

RNF12 regulates specific gene expression programmes involved in X-chromosome inactivation (Jonkers et al, 2009; Shin et al, 2010; Barakat et al, 2011; Gontan et al, 2012, 2018), neurodevelopment (Bustos et al, 2020), and gametogenesis (Segarra-Fas et al, 2022). This occurs largely via ubiquitylation and resulting proteasomal degradation of the transcriptional regulator ZFP42/REX1 (Gontan et al, 2012; Bustos et al, 2020; Segarra-Fas et al, 2022). Beyond this, the molecular details of how RNF12 controls specific gene expression signatures remain unclear. RNF12 directly interacts with REX1, via sequences in both the N- and C-terminal regions of the RNF12 polypeptide (Gontan et al, 2012; Bustos et al, 2020). However, it is not known whether RNF12 engages REX1 specifically on chromatin and/or at specific sites, such as transcriptionally active promoters. Furthermore, the broader role that chromatin context plays in regulation of RNF12 activity towards REX1 has not been studied.

Here, we show that chromatin forms a platform that is required for efficient RNF12 substrate ubiquitylation and transcriptional regulation. We find using proximity labelling that RNF12 engages

[1]MRC Protein Phosphorylation and Ubiquitylation Unit, School of Life Sciences, University of Dundee, Dundee, UK [2]Department of Developmental Biology, Erasmus University Medical Center, Rotterdam, Netherlands [3]Enabling Technologies Group, Sanford Research, Sioux Falls, SD, USA [4]Department of Pediatrics, Sanford School of Medicine, University of South Dakota, Sioux Falls, SD, USA [5]Proteomics Center and Department of Biochemistry, Erasmus University Medical Center, Rotterdam, Netherlands [6]Department of Biochemistry and Molecular Biology, University of Arkansas for Medical Sciences, Little Rock, AR, USA [7]Pediatrics and Rare Diseases Group, Sanford Research, Sioux Falls, SD, USA

Correspondence: g.m.findlay@dundee.ac.uk

chromatin components, and chromatin immunoprecipitation (ChIP) followed by DNA sequencing (ChIP-seq) analyses reveal that RNF12 is recruited to specific chromatin regions. In particular, RNF12 is recruited along with its REX1 substrate at target gene promoters, leading to REX1 ubiquitylation and gene regulation. Mechanistically, RNF12 engages chromatin via a conserved basic region adjacent to the RING domain, which is critical for efficient REX1 binding and ubiquitylation and RNF12-dependent gene regulation. Furthermore, RNF12 N-terminal sequences suppress chromatin recruitment and substrate ubiquitylation, uncovering a previously unappreciated autoinhibitory mechanism. Taken together, our results provide insight into mechanisms by which sequence-specific chromatin targeting of an E3 ubiquitin ligase coordinates catalytic activity with transcription factor substrate ubiquitylation, enabling implementation of specific gene expression programmes.

# Results

### RNF12/RLIM proximity-induced labelling mass spectrometry identifies REX1 substrate and other chromatin associated components

A key function of RNF12 is ubiquitylation and resulting proteasomal degradation of developmental transcriptional regulators (Ostendorff et al, 2002; Her & Chung, 2009; Gontan et al, 2012; Zhang et al, 2012; Gao et al, 2016; Wang et al, 2019). Chief among these is the transcription factor ZFP42/REX1 (Gontan et al, 2012, 2018), which patterns developmental gene expression during embryonic stem cell differentiation and is associated with developmental abnormalities (Bustos et al, 2020; Segarra-Fas et al, 2022). However, the mechanisms by which RNF12 is targeted to substrates such as REX1 to control gene expression remain unclear, but are critical to understand RNF12 regulation and function in normal and disease states.

To address this question, we took a proximity ligation approach to identify RNF12 proximal proteins. TurboID labelling (Branon et al, 2018) is a proximity ligation-based method that tethers a promiscuous biotin ligase to a protein of interest to rapidly biotinylate and identify proximal proteins. Thus, we fused TurboID machinery to the RNF12 N-terminus to identify proteins that are specifically labelled by RNF12 proximity. Expression of RNF12 TurboID and TurboID machinery alone was induced in mouse embryonic stem cells (mESCs), incubated with biotin to induce proximity labelling, and treated with the proteasome inhibitor MG132 to stabilise RNF12 itself and proximal proteins that might otherwise be targeted for proteasomal degradation. Correct expression and nuclear localisation of HA-TurboID RNF12 were confirmed by immunoblotting (Fig 1A) and immunofluorescence (Fig 1B). RNF12 proximal proteins were then identified by streptavidin pull-down and mass spectrometry, and peptides and proteins were quantified to determine fold-change and statistical significance. Proteins whose labelling is increased >twofold in RNF12 TurboID samples relative to TurboID control were pinpointed (285 proteins; Fig 1C and Table S1); proof of principle for this utility of this approach to identify RNF12 proximal proteins was provided by identification of known substrate REX1 (Fig 1C).

Next, we interrogated priority RNF12 proximity-labelled proteins for further information about RNF12 regulation and/or function. As RNF12 is localised to the nucleus (Fig 1B) (Jiao et al, 2013; Bustos et al, 2020), proteins with annotated nuclear localisation and/or function were prioritised from the >twofold enriched cohort (132 proteins; Table S2). We then performed Database for Annotation, Visualization, and Integrated Discovery (DAVID) functional enrichment analysis (Huang et al, 2009; Sherman et al, 2022), which incorporates many annotation terms including gene ontology (GO) and is more practical for analysis of focussed datasets in comparison with gene set enrichment analysis. DAVID indicates that nuclear RNF12 proximity-labelled proteins are significantly enriched for chromatin-specific functions, such as DNA damage response, regulation of gene expression, and DNA replication (Fig 1D). We also examined all statistically significant RNF12 proximal proteins identified by TurboID ($P < 0.05$). Although the dataset is too small to determine significantly enriched functions, several chromatin-associated factors are identified (highlighted in yellow) (Table S3).

We also compared our findings from RNF12 TurboID with previously published findings from RNF12 affinity-purification mass spectrometry (AP-MS) (Gontan et al, 2012) (Table S4). DAVID functional enrichment analysis indicates that RNF12-interacting proteins identified by AP-MS are similarly enriched for chromatin-specific functions (Table S5). Only six proteins including known substrate REX1 were identified by both RNF12 TurboID and AP-MS (Fig 1E), presumably as a result of different technical approaches (AP-MS identifies stable interactions versus TurboID identification of proximal proteins) and biological contexts (AP-MS was performed on female mESCs differentiated for 3 d, whereas TurboID was performed on pluripotent male mESCs). However, of the common proteins, PCNA, SMU1, and WRNIP1 are chromatin associated (Fig 1E). In summary, our data strongly suggest that the chromatin environment forms a key component of RNF12 regulation and function, consistent with our previous data indicating that RNF12 is recruited to chromatin in mESCs (Segarra-Fas et al, 2022).

### RNF12 engages chromatin

Considering these findings, we explored the biological function of RNF12 chromatin recruitment. Using biochemical fractionation, we determined that RNF12 is present in both soluble cytoplasm/nucleoplasm and on chromatin alongside REX1 (Fig 2A). Effective separation of chromatin from other soluble nuclear/cytoplasmic material was confirmed by immunoblotting for $\beta$III-tubulin (TUB$\beta$3), a component of microtubules, and Histone H3 phospho-Ser10 (pH3), a core component of chromatin (Fig 2A). Quantification of the relative amounts of RNF12 and REX1 found on chromatin compared with the soluble cellular fraction indicates that a significant proportion of RNF12 (25.7% ± 11.4%) and REX1 (52.4% ± 18.8%) is recruited to chromatin (Fig 2B and C). As RNF12 (Jiao et al, 2013; Bustos et al, 2020) and REX1 (Gontan et al, 2012) are both localised to the nucleus, the remainder is most likely present in the nucleoplasm and/or other nuclear structures. These data therefore indicate that RNF12 is recruited to chromatin along with key substrate REX1, although the majority of RNF12 and a significant proportion of REX1 is found in other nuclear compartments.

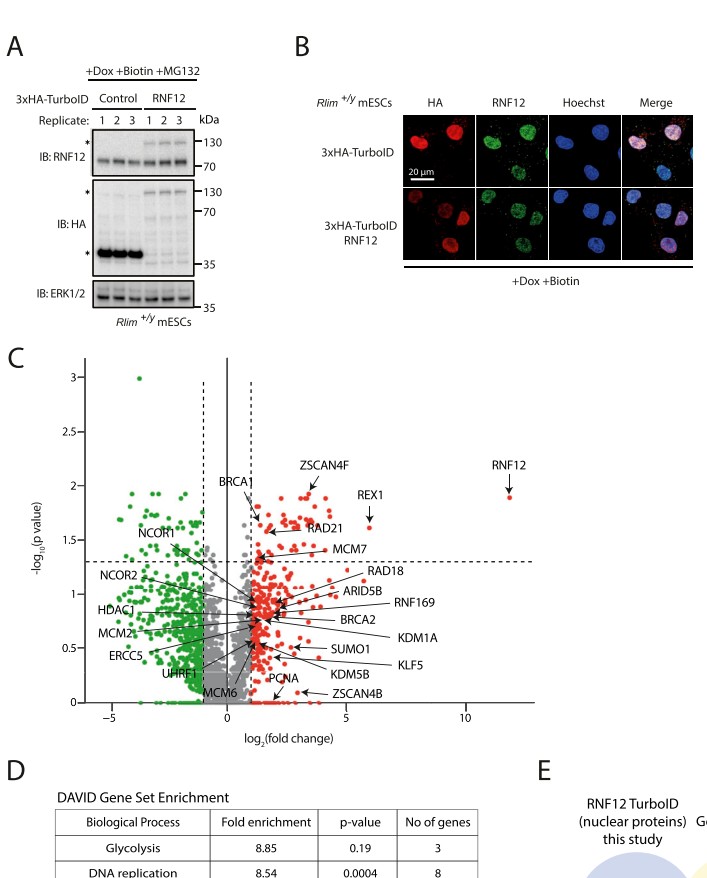

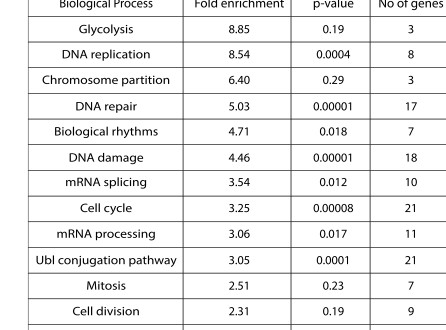

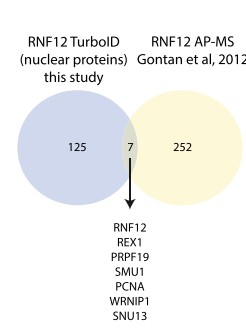

**Figure 1. RNF12 TurboID proximity labelling identifies chromatin-associated proteins.**
**(A)** *Rlim*⁺/ʸ mouse embryonic stem cells (mESCs) stably overexpressing HA-TurboID RNF12 and HA-TurboID control were treated with MG132, doxycycline, and biotin in triplicate. Levels of HA-TurboID RNF12 and HA-TurboID control were determined by immunoblotting and indicated by an asterisk. ERK1/2 is shown as a loading control. **(B)** Immunofluorescence analysis of doxycycline and biotin-treated *Rlim*⁺/ʸ mESCs stably overexpressing HA-TurboID RNF12 and HA-TurboID control. HA, total RNF12, and Hoechst as a nuclear stain are shown. **(C)** Volcano plot showing relative change in protein abundance of biotinylated proteins comparing MG132, doxycycline, and biotin-treated *Rlim*⁺/ʸ mESCs stably overexpressing HA-TurboID RNF12 to HA-TurboID control. Red data points indicate proteins displaying a >twofold increase in intensity in HA-TurboID RNF12-expressing mESCs. **(D)** Database for Annotation, Visualization, and Integrated Discovery analysis showing enriched biological processes within the gene set encoding proteins with annotated nuclear localisation and/or function and which display >twofold increased intensity in HA-TurboID RNF12-overexpressing cells compared with control. **(E)** Venn diagram displaying the number of proteins identified to have >twofold increase in intensity in HA-TurboID RNF12-overexpressing cells relative to control, compared with the number of RNF12-interacting proteins identified by affinity-purification mass spectrometry (Gontan et al, 2012). Proteins common to both datasets are indicated. Source data are available for this figure.

**DAVID Gene Set Enrichment**

| Biological Process | Fold enrichment | p-value | No of genes |
|---|---|---|---|
| Glycolysis | 8.85 | 0.19 | 3 |
| DNA replication | 8.54 | 0.0004 | 8 |
| Chromosome partition | 6.40 | 0.29 | 3 |
| DNA repair | 5.03 | 0.00001 | 17 |
| Biological rhythms | 4.71 | 0.018 | 7 |
| DNA damage | 4.46 | 0.00001 | 18 |
| mRNA splicing | 3.54 | 0.012 | 10 |
| Cell cycle | 3.25 | 0.00008 | 21 |
| mRNA processing | 3.06 | 0.017 | 11 |
| Ubl conjugation pathway | 3.05 | 0.0001 | 21 |
| Mitosis | 2.51 | 0.23 | 7 |
| Cell division | 2.31 | 0.19 | 9 |
| Transcription regulation | 1.94 | 0.0003 | 37 |
| Transcription | 1.89 | 0.0006 | 37 |

## RNF12 and REX1 substrate are largely co-localised to specific gene regulatory regions

As RNF12 and REX1 are located on chromatin, we next investigated the genomic regions occupied by these proteins. REX1 genome occupancy in mESCs has been determined previously by chromatin immunoprecipitation followed by DNA sequencing (ChIP-seq) (Gontan et al, 2012), which prompted us to perform RNF12 ChIP-seq to investigate whether RNF12 and REX1 are recruited to specific and/or common locations. Undifferentiated female Rnf12⁺/⁻ mESCs treated with the proteasome inhibitor MG132 and expressing FLAG–V5-RNF12^WT or FLAG–V5-RNF12^H569A,C572A, a catalytically inactive mutant of RNF12 that likely disrupts folding of the catalytic RING domain, were used for ChIP-seq analyses to determine RNF12 genome occupancy.

We first addressed whether RNF12 genome occupancy overlaps with that of REX1 in mESCs. Overlap analysis of significant peak regions from each ChIP-seq dataset suggests that RNF12^WT, RNF12^H569A,C572A, and REX1 are recruited to both unique and shared chromatin regions (Fig S1A). As the degree of overlap is highly dependent on the thresholds used during peak calling, we used MAnorm (Shao et al, 2012) to quantitatively compare the signal at peaks, enabling the identification of common and shared peak regions. This quantitative peak analysis reveals that RNF12^WT, RNF12^H569A,C572A, and REX1 are largely recruited to shared genome sequences, with a small sub-set of genomic regions occupied by either RNF12 or REX1 alone (Figs 2D and S1B). Indeed, correlation analysis suggests a strong correlation between RNF12 and REX1 sites of genome occupancy (Fig S1C). Therefore, our data suggest that RNF12 and REX1 largely occupy common sites within the

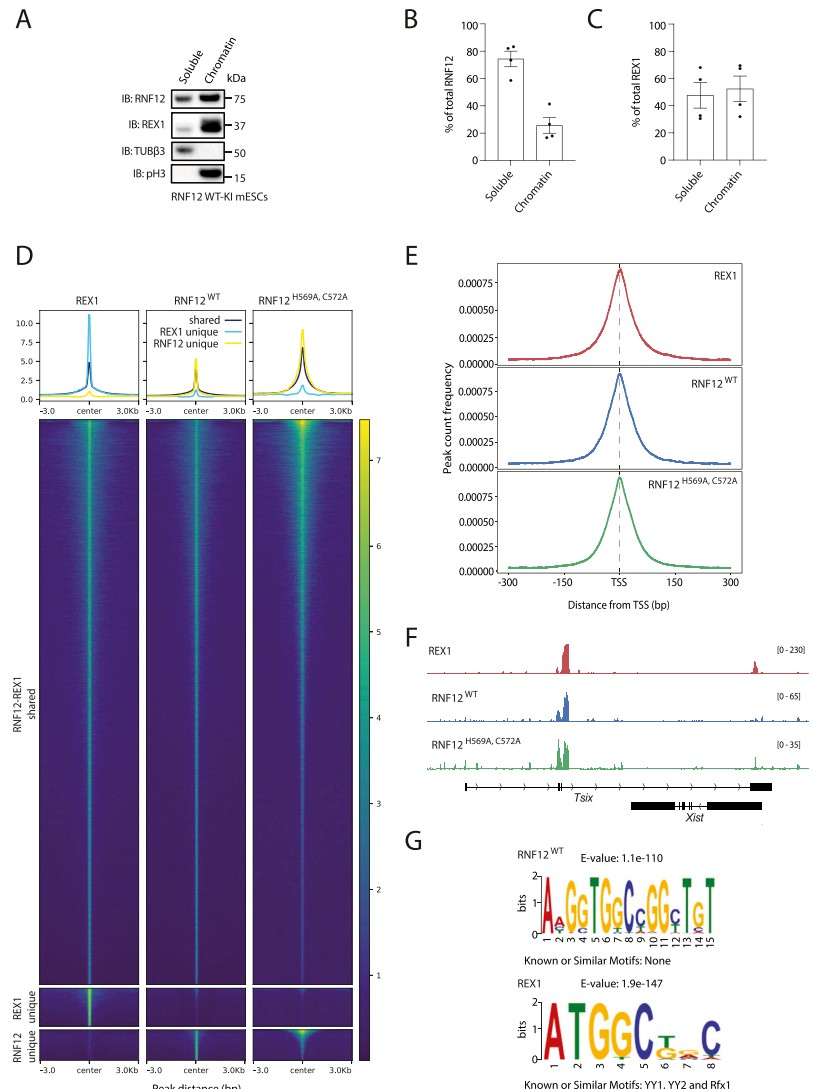

**Figure 2. RNF12 colocalises with REX1 transcription factor substrate at specific genomic regions.**
**(A)** RNF12 WT knock-in (WT-KI) mouse embryonic stem cells (mESCs) were subjected to chromatin fractionation, and RNF12, REX1, $\beta$III-tubulin (TUB$\beta$3), and phospho-Ser10 Histone H3 (pH3) levels determined by immunoblotting. TUB$\beta$3 is used as a marker of the soluble fraction, and pH3 as a marker of the chromatin fraction. Data are representative of n = 4 independent experiments. **(B)** Quantification of the RNF12 signal observed in (A). Data are represented as the proportion of RNF12 in soluble and chromatin fractions. As the protein content of the soluble fraction is higher than that of the chromatin fraction, the relative proportion of protein in soluble versus chromatin fraction was calculated in all subsequent quantifications. Data represented as mean ± SEM (n = 4). **(C)** Quantification of the REX1 signal observed in (A). Data represented as mean ± SEM (n = 4). **(D)** Heatmap showing the enrichment of RNF12$^{WT}$, RNF12$^{H569A,C572A}$, and REX1 ChIP-seq data at REX1-RNF12 shared peaks and unique peaks identified by MAnorm. The signal in the region ±3 kb of the peak center is shown in the heatmap and summarized in the profile plot above. The colour bar shows the Poisson $P$-value ($-\log_{10}$) calculated by MACS2 using control as lambda and treatment as observation, whereas the y-axis of the profile plot shows the mean Poisson $P$-value across all peaks of each category. **(E)** Peak count frequency relative to distance from transcriptional start sites of ChIP-seq peaks identified for RNF12$^{WT}$, RNF12$^{H569A,C572A}$, and REX1. **(F)** Genome browser view of the input-normalized tracks for RNF12$^{WT}$, RNF12$^{H569A,C572A}$, and REX1 at the *Xist*/*Tsix* locus. The y-axes show the $-\log_{10}$(Poisson $P$-value) as described in (D). **(G)** DNA sequence motif enrichment analysis of ChIP-seq sequences identified for RNF12$^{WT}$ and REX1 (top hit shown).
Source data are available for this figure.

genome, in addition to a sub-set of genomic regions that are uniquely bound by either RNF12 or REX1.

We next explored the nature of the specific genomic regions occupied by RNF12 and REX1. Feature distribution analysis of RNF12$^{WT}$, RNF12$^{H569A,C572A}$, and REX1-bound genomic regions indicates enrichment of promoter proximal regions (Fig S1D). Furthermore, analysis of the relative positions of RNF12$^{WT}$, RNF12$^{H569A,C572A}$, and REX1 chromatin binding sites indicates strong enrichment of RNF12 and REX1 at transcription start sites (Fig 2E). As REX1 was previously shown to be enriched close to transcription start sites (Gontan et al, 2012), this is consistent with the overlap observed between RNF12 and REX1 chromatin binding sites. As an example, the region encompassing the long non-coding RNA *Xist* and its antisense transcript *Tsix*, which are located in the X inactivation center and play crucial roles in the regulation of X chromosome inactivation (Jonkers et al, 2009; Barakat et al, 2011; Gontan et al, 2012, 2018; Wang et al, 2017), exhibits RNF12 and REX1 peaks of genome occupancy (Fig 2F).

Our findings suggest that RNF12 and REX1 largely occupy common genomic locations. However, the existence of genomic regions that are occupied by RNF12 and REX1 alone suggests distinct specificities for chromatin recruitment. We thus sought to determine the genome sequence motifs occupied by RNF12 and REX1. Analysis of REX1 genomic binding sites suggests enrichment of a consensus motif previously associated with the REX1/YY1/YY2 family of transcriptional regulators as expected (Fig 2G) (Kim et al, 2007). Analysis of RNF12 recruitment sites, which largely overlap with REX1 recruitment sites (Fig 2D), nevertheless reveals a distinct sequence recruitment motif (Fig 2G). This is consistent with the existence of a sub-set of promoters engaged by either RNF12 or REX1 alone (Fig 2D), suggesting that these genes may be regulated by RNF12 or REX1 independently of the RNF12-REX1 axis. Indeed, we find that gene loci bound by either RNF12 or REX1 alone exhibit distinct predicted transcription factor binding profiles (Fig S1E), suggesting that different families of transcription factors may participate in regulation of these genes. Taken together, our findings indicate that although RNF12 and REX1

are largely recruited to shared genomic sites, this may occur via distinct sequence motifs, potentially enabling a sub-set of genes to be regulated by RNF12 via other transcription factors independently of the core RNF12–REX1 axis.

## RNF12 substrate REX1 is efficiently ubiquitylated specifically on chromatin

Our demonstration that RNF12 and REX1 are co-localised to specific gene regulatory regions prompts the hypothesis that chromatin recruitment is a key event to enable RNF12 ubiquitylation of REX1 at specific genomic locations to regulate gene expression. Therefore, we measured REX1 ubiquitylation on chromatin and in other cellular compartments by stabilising ubiquitylated REX1 using the proteasome inhibitor MG132, performing chromatin fractionation, and specifically quantifying REX1 ubiquitylated species, which are distinguished as a series of distinct bands migrating at higher molecular weight than unmodified REX1 (Fig 3A and B). This analysis suggests that endogenous REX1 is heavily ubiquitylated in the chromatin fraction compared with other cellular compartments (Fig 3A and B). As expected, REX1 ubiquitylation is reduced in RNF12-deficient ($Rlim^{-/y}$) mESCs, although residual REX1 ubiquitylation is observed, particularly shorter ubiquitin chains (Fig 3C). Quantification confirms that REX1 ubiquitylation is reduced in RNF12-deficient mESCs (Fig 3D). RNF12-dependent REX1 ubiquitylation on chromatin is also increased in RNF12-deficient mESCs reconstituted with RNF12$^{WT}$, but not with catalytic-deficient mutants of RNF12 (RNF12$^{W576Y}$ and RNF12$^{H569A,C572A}$) (Fig 3E and F), indicating that REX1 ubiquitylation on chromatin requires RNF12 catalytic activity. Notably, two distinct RNF12 catalytic mutants have a similar impact on chromatin recruitment and REX1 ubiquitylation; RNF12$^{W576Y}$, which impairs interaction of RNF12 with E2 conjugating enzymes (Bustos et al, 2018) and RNF12$^{H569A,C572A}$, which likely disrupts the folding of the RING domain.

## RNF12 is recruited to chromatin via the basic region (BR)

As most of the RNF12-dependent REX1 ubiquitylation takes place on chromatin, we next sought to address the mechanism by which RNF12 engages chromatin. To this end, we performed deletion mutagenesis to identify RNF12 regions that are required for chromatin recruitment (Fig 4A). As expected, deletion of the RNF12 nuclear localisation signal (NLS) reduces chromatin recruitment (Fig 4B and C), presumably via effects on RNF12 nuclear localisation. In contrast, deletion of the RNF12 nuclear export signal (NES) or the catalytic RING domain has no discernible impact on chromatin recruitment (Fig 4B and C). However, deletion of a conserved (Fig S2A) basic region (BR) prevents recruitment to chromatin (Fig 4B and C). We confirm that RNF12 deletion constructs are correctly expressed in total cell extracts (Fig S2B) and, as shown previously (Segarra-Fas et al, 2022), localise to the nucleus as expected (Fig S2C). Furthermore, the isolated RNF12 BR is effectively recruited to chromatin (Fig 4D and E) and correctly expressed in total cell extracts (Fig S2D), indicating that the BR is both necessary and sufficient for RNF12 to engage chromatin. Interestingly, deletion of RNF12 N-terminal sequences drives increased chromatin

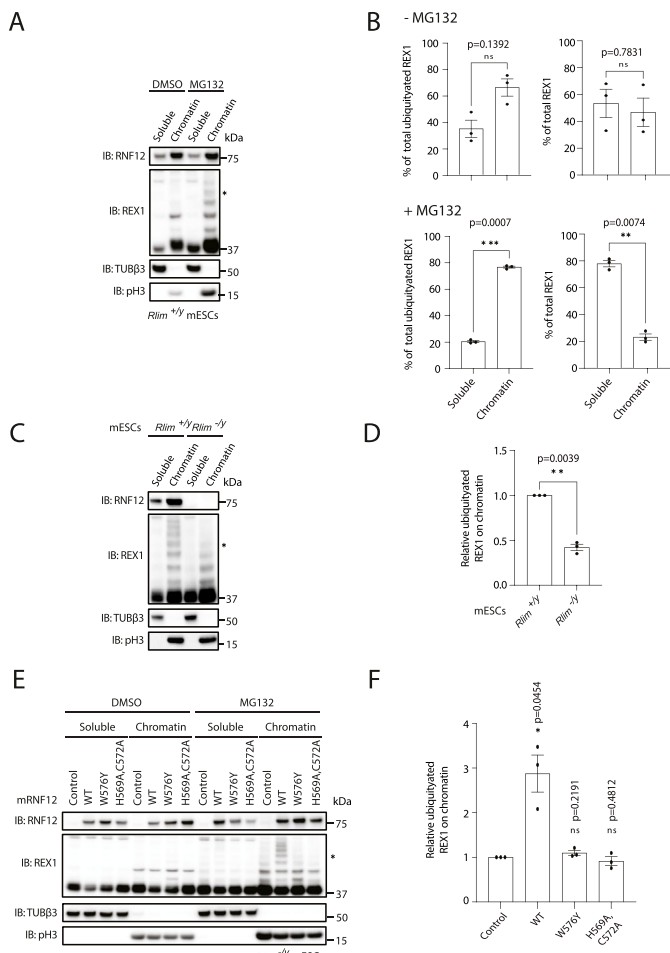

**Figure 3. RNF12 substrate REX1 is efficiently ubiquitylated on chromatin.**
**(A)** $Rlim^{+/y}$ mouse embryonic stem cells (mESCs) were treated with DMSO or MG132 inhibitor for 1 h to stabilise ubiquitylated REX1 and subjected to chromatin fractionation. RNF12, REX1, $\beta$III-tubulin (TUB$\beta$3), and phospho-Ser10 Histone H3 (pH3) levels were determined by immunoblotting. Data are representative of n = 3 independent experiments. **(B)** REX1 ubiquitylation in soluble and chromatin fractions of $Rlim^{+/y}$ mESCs from (A) was quantified by determining relative average intensity of the fourth ubiquitylated band (REX1-Ub$^4$; indicated by an asterisk in (A)) and normalizing to total REX1 levels. Data represented as mean ± SEM (n = 3). Statistical significance was determined by paired $t$ test; two-sided, confidence level 95%. **(C)** $Rlim^{+/y}$ and RNF12 knock-out ($Rlim^{-/y}$) mESCs were treated with MG132 inhibitor for 1 h subjected to chromatin fractionation. RNF12, REX1, $\beta$III-tubulin (TUB$\beta$3), and phospho-Ser10 Histone H3 (pH3) levels were determined by immunoblotting. Data are representative of n = 3 independent experiments. **(D)** REX1 ubiquitylation in the chromatin fraction of $Rlim^{+/y}$ and $Rlim^{-/y}$ mESCs from (C) was quantified by determining relative average intensity of the fourth ubiquitylated band (REX1-Ub$^4$; indicated by an asterisk in (C)) and normalizing to total REX1 levels. Data represented as mean ± SEM (n = 3). Statistical significance was determined by paired $t$ test; two-sided, confidence level 95%. **(E)** $Rlim^{-/y}$ mESCs expressing either empty vector (control), mouse RNF12$^{WT}$, RNF12$^{W576Y}$, and RNF12$^{H569A,C572A}$ were treated with DMSO or MG132 for 1 h and subjected to chromatin fractionation. RNF12, REX1, $\beta$III-tubulin (TUB$\beta$3), and phospho-Ser10 Histone H3 (pH3) levels were determined by immunoblotting. Data are representative of n = 3 independent experiments. **(F)** REX1 ubiquitylation in the chromatin fraction from (E) was quantified by determining relative average intensity of the fourth ubiquitylated band (REX1-Ub$^4$; indicated by an asterisk in (E)) and normalizing to total REX1 levels. Data represented as mean ± SEM (n = 3). Statistical significance was determined by paired $t$ test; two-sided, confidence level 95%.
Source data are available for this figure.

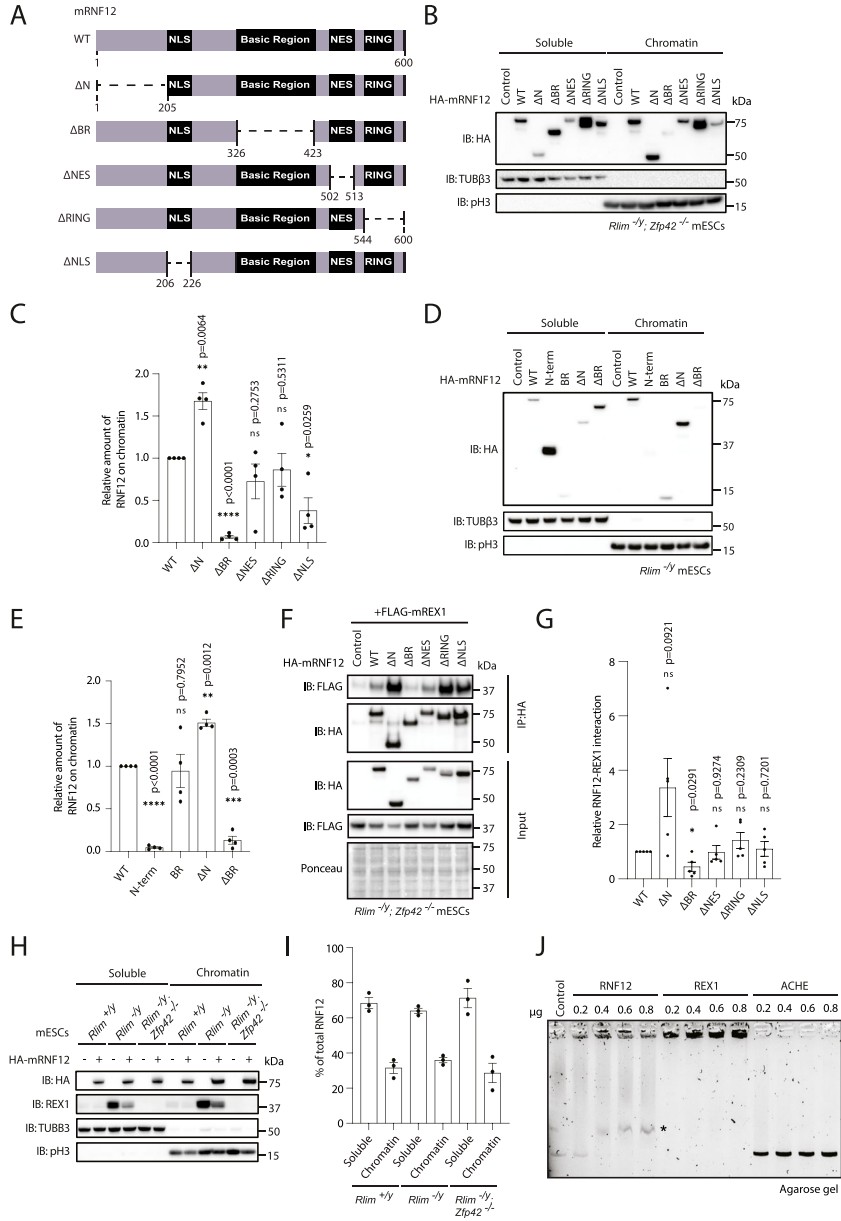

**Figure 4. RNF12 chromatin recruitment is largely REX1 independent.**
**(A)** Schematic of the structure of mouse RNF12$^{WT}$ and deletion mutants. Indicated are the amino acid boundaries of each deletion. **(B)** RNF12/REX1 double knock-out ($Rlim^{-/y}$; $Zfp42^{-/-}$) mouse embryonic stem cells (mESCs) expressing FLAG-REX1 with either empty vector (control), HA-tagged mouse RNF12$^{WT}$ (1–600), RNF12$^{\Delta1-206}$ (ΔN), RNF12$^{\Delta326-423}$ (ΔBR), RNF12$^{\Delta502-513}$ (ΔNES), RNF12$^{\Delta543-600}$ (ΔRING), or RNF12$^{\Delta206-226}$ (ΔNLS) were subjected to chromatin fractionation. HA-RNF12, βIII-tubulin (TUBβ3) and phospho-Ser10 Histone H3 (pH3) levels were determined by immunoblotting. Data are representative of n = 4 independent experiments. **(C)** Quantification of HA-RNF12 deletion mutant protein levels observed in the chromatin fraction in (B) expressed relative to RNF12$^{WT}$. Data represented as mean ± SEM (n = 4). Statistical significance was determined by paired $t$ test; two-sided, confidence level 95%. **(D)** $Rlim^{-/y}$ mESCs expressing either empty vector (control), HA-tagged mouse RNF12$^{WT}$ (1–600), RNF12$^{1-205}$ (N-term) RNF12$^{326-423}$ (basic region) RNF12$^{\Delta1-206}$ (ΔN), and RNF12$^{\Delta326-423}$ (ΔBR) were subjected to chromatin fractionation. HA-RNF12, βIII-tubulin (TUBβ3), and phospho-Ser10 Histone H3 (pH3) levels were determined by immunoblotting. Data are representative of n = 4 independent experiments. **(E)** Quantification of HA-RNF12 deletion mutant protein levels observed in the chromatin fraction in (D) expressed relative to RNF12$^{WT}$. Data represented as mean ± SEM (n = 4). Statistical significance was determined by paired $t$ test; two-sided, confidence level 95%. **(F)** RNF12/REX1 double knock-out ($Rlim^{-/y}$; $Zfp42^{-/-}$) mESCs expressing FLAG-REX1 with either empty vector (control), HA-tagged mouse RNF12$^{WT}$, or the indicated HA-RNF12 deletion mutants were treated with MG132 for 2 h and HA-RNF12 immunoprecipitated using HA resin. HA-RNF12 and FLAG-REX1 levels were determined by immunoblotting, and Ponceau S staining is shown as a loading control. Data are representative of n = 5 independent experiments. **(G)** Quantification of data from (F) represented as mean ± SEM (n = 5). Statistical significance was determined by paired $t$ test; two-sided, confidence level 95%. **(H)** Control ($Rlim^{+/y}$), RNF12 knock-out ($Rlim^{-/y}$), and RNF12/REX1 double knock-out ($Rlim^{-/y}$; $Zfp42^{-/-}$) mESCs expressing either empty vector (−) or HA-tagged mouse RNF12$^{WT}$ were subjected to chromatin fractionation. HA-RNF12, REX1, βIII-tubulin (TUBβ3), and phospho-Ser10 Histone H3 (pH3) levels were determined by immunoblotting. Data are representative of n = 3 independent experiments. **(I)** Quantification of HA-tagged mouse RNF12$^{WT}$ protein levels observed in the chromatin fraction in (H) expressed as a percentage of total HA-RNF12. Data represented as mean ± SEM (n = 3). **(J)** Electrophoretic mobility shift analysis of linearized pCAGGS plasmid DNA (0.5 μg) incubated with increasing concentrations (0.2–0.8 μg) of RNF12, REX1, and ACHE recombinant proteins and analysed on an 0.8% agarose gel. Data are representative of n = 3 independent experiments. Source data are available for this figure.

recruitment (Fig 4B and C), suggesting that the RNF12 N-terminus functions to inhibit RNF12 chromatin engagement. Indeed, unlike the RNF12 BR, the isolated N-terminal region is not recruited to chromatin (Fig 4D and E), despite being present in the nucleus (Fig S2E). Taken together, our results reveal the RNF12 BR as a major determinant of RNF12 chromatin recruitment.

## RNF12 chromatin recruitment mechanism is REX1 independent

Previous work has implicated the RNF12 BR, amongst other regions, in REX1 interaction (Gontan et al, 2012), suggesting that substrate engagement could be a key mechanism for chromatin recruitment.

We first confirmed that the RNF12 BR is required for interaction with REX1 substrate. In immunoprecipitation assays, RNF12 interacts with REX1, and this is reduced by deletion of the RNF12 BR (Fig 4F and G), confirming the role of the RNF12 BR in REX1 substrate interaction. Interestingly, RNF12 N-terminal deletion leads to increased REX1 binding (Fig 4F and G), suggesting that the RNF12 N-terminus not only inhibits chromatin recruitment but also REX1 substrate interaction.

Considering our findings that RNF12 and REX1 largely occupy common genomic sequences and that the RNF12 basic and N-terminal regions modulate both chromatin recruitment and REX1 interaction, we next tested whether REX1 engagement is the mechanism by which

RNF12 is recruited to chromatin. To this end, we took advantage of an allelic series of WT, RNF12-deficient ($Rlim^{-/y}$), and RNF12/REX1-deficient ($Rlim^{-/y}$; $Zfp42^{-/-}$) mESC lines reconstituted with HA-RNF12$^{WT}$. As in control cells, HA-RNF12$^{WT}$ is efficiently recruited to chromatin in either RNF12-deficient or RNF12/REX1-deficient mESCs (Fig 4H and I), suggesting that interaction with REX1 is not a major mechanism for RNF12 chromatin recruitment. Consistent with this notion, recruitment of RNF12 BR and N-terminal deletion mutants to chromatin is not altered by REX1 deletion (Fig S2F and G).

Our data indicate that RNF12 chromatin engagement largely occurs independent of REX1 interaction. As RNF12 chromatin recruitment is mediated by the BR, we next asked whether this positively charged region might mediate direct electrostatic interactions with negatively charged DNA. To test this, we incubated recombinant RNF12 with circular plasmid DNA (pCAGGS) and performed electrophoretic mobility shift analysis (EMSA). In the absence of protein or in the presence of a negative control protein ACHE that does not bind DNA, pCAGGS plasmid is resolved at the expected molecular weight by agarose gel electrophoresis (Fig 4J). However, addition of the REX1 transcription factor, which directly binds DNA, reduces the electrophoretic mobility of plasmid DNA upon EMSA (Fig 4J). Similarly, RNF12 reduces the electrophoretic mobility of plasmid DNA upon EMSA (Fig 4J, see asterisk), suggesting that RNF12 also has the capacity to directly interact with DNA. Taken together, our results indicate that the RNF12 BR mediates recruitment to chromatin in a manner that is independent of REX1, potentially by directly interacting with DNA.

## RNF12 chromatin recruitment via the basic region is required for substrate processing

We next sought to determine the specific sequences within the RNF12 BR that are required for chromatin recruitment. To this end, we generated three smaller BR deletions (ΔBR1 lacking amino acids 326–348, ΔBR2 lacking amino acids 349–381 and ΔBR3 lacking amino acids 382–423) (Fig 5A) and addressed the impact of these sequences on chromatin recruitment. As shown previously, deletion of the RNF12 BR abolishes chromatin recruitment (Fig 5B and C). Similarly, deletion of BR1 and BR2 disrupts chromatin recruitment, although to a lesser extent (Fig 5B and C). In contrast, deletion of BR3 increases RNF12 chromatin recruitment (Fig 5B and C). This region has fewer basic residues than BR1 and BR2, suggesting that BR3 is not only dispensable for chromatin engagement but may encode an element that autoinhibits engagement of chromatin by RNF12.

As RNF12 BR deletions can, in principle, impact catalytic activity (Bustos et al, 2018), REX1 substrate recruitment (Gontan et al, 2012) (Fig 4F and G), and/or chromatin recruitment (Fig 4B and C), we set out to distinguish these possibilities. To this end, we measured catalytic activity of RNF12 BR deletion mutants in the presence of recombinant ubiquitin, UBE2D1 (E2), UBE1 (E1), and REX1 substrate in vitro. As shown previously, RNF12$^{WT}$ catalyses REX1 substrate ubiquitylation (Bustos et al, 2018) (Fig 5D and E). Interestingly, deletion of BR, BR1, or BR3 significantly decreases REX1 ubiquitylation (Fig 5D and E), although RNF12 catalytic activity towards REX1 is largely unaffected by BR2 deletion (Fig 5D and E). As expected, engagement of REX1 by RNF12 is also largely unaffected by BR2 deletion, when compared with deletion of the BR (Fig 5F). These data indicate that

whereas the RNF12 BR performs functions that are required for chromatin recruitment, substrate engagement, and catalysis, specific deletion of the RNF12 BR2 region partially separates these functions by impacting primarily on chromatin recruitment, without significantly impacting on catalytic activity and substrate engagement.

We then explored the effect of RNF12 BR deletions on REX1 substrate ubiquitylation in mESCs. Using MG132 treatment in combination with chromatin fractionation as before, we were able to sensitively measure RNF12-dependent REX1 ubiquitylation (Fig 5G and H). Consistent with the impact on chromatin recruitment, the RNF12 BR is required for efficient REX1 ubiquitylation (Fig 5G and H). However, RNF12 BR1 and BR3 deletions drive efficient REX1 ubiquitylation (Fig 5G and H), despite differing relative impacts on chromatin recruitment (Fig 5B and C). Although REX1 ubiquitylation is observed with the RNF12 BR1 deletion, this appears to be a consequence of increased chromatin recruitment in the presence of MG132, when compared with that observed for RNF12 BR2 and BR3 deletions (Fig 5G). In contrast, RNF12 BR2 deletion is impaired for both REX1 ubiquitylation (Fig 5G and H) and chromatin recruitment (Fig 5B and C), suggesting that the BR2 region plays a critical role in RNF12 chromatin recruitment, which in turn impacts substrate ubiquitylation. This notion is supported by REX1 stability, which is more profoundly affected by RNF12 BR and BR2 deletion, when compared with BR1 and BR3 deletion (Fig 5I and J). However, in contrast to RNF12 BR deletion, the RNF12 BR2 deletion mutant undergoes efficient degradation mediated by autoubiquitylation (Fig 5I and J), consistent with our observation that RNF12 BR2 deletion does not have a major impact on catalytic activity per se (Fig 5D). Therefore, these data support the conclusion that the RNF12 BR is critical for REX1 substrate ubiquitylation by enabling RNF12 recruitment to chromatin.

## RNF12 N-terminal region negatively regulates chromatin recruitment and substrate ubiquitylation

We have demonstrated that the RNF12 BR is required for chromatin recruitment, substrate engagement, and ubiquitylation. However, we observe an opposing effect of the RNF12 N-terminal region, deletion of which leads to increased RNF12 chromatin association, suggesting that the RNF12 N-terminus somehow acts to suppress chromatin recruitment. This prompted us to address the functional importance of the RNF12 N-terminal region for substrate ubiquitylation.

First, we sought to define the specific sequences within the RNF12 N-terminal region that are required to modulate chromatin recruitment. To this end, we generated three smaller deletions of the RNF12 N-terminus (ΔN1 lacking amino acids 1–68, ΔN2 lacking amino acids 69–135 and ΔN3 lacking amino acids 136–206) and determined the impact of these N-terminal sequences on chromatin recruitment (Fig 6A). As shown previously, deletion of the RNF12 N-terminus augments chromatin recruitment (Fig 4B and C). However, deletion of N1, N2, and N3 individually has no significant impact on RNF12 chromatin recruitment (Fig 6B and C), and these mutants behave similarly to RNF12$^{WT}$. These data suggest that the entire RNF12 N-terminal region (amino acids 1–206) is required to negatively regulate chromatin recruitment.

We then explored the functional impact of RNF12 N-terminal sequences on REX1 substrate ubiquitylation. Using MG132 treatment

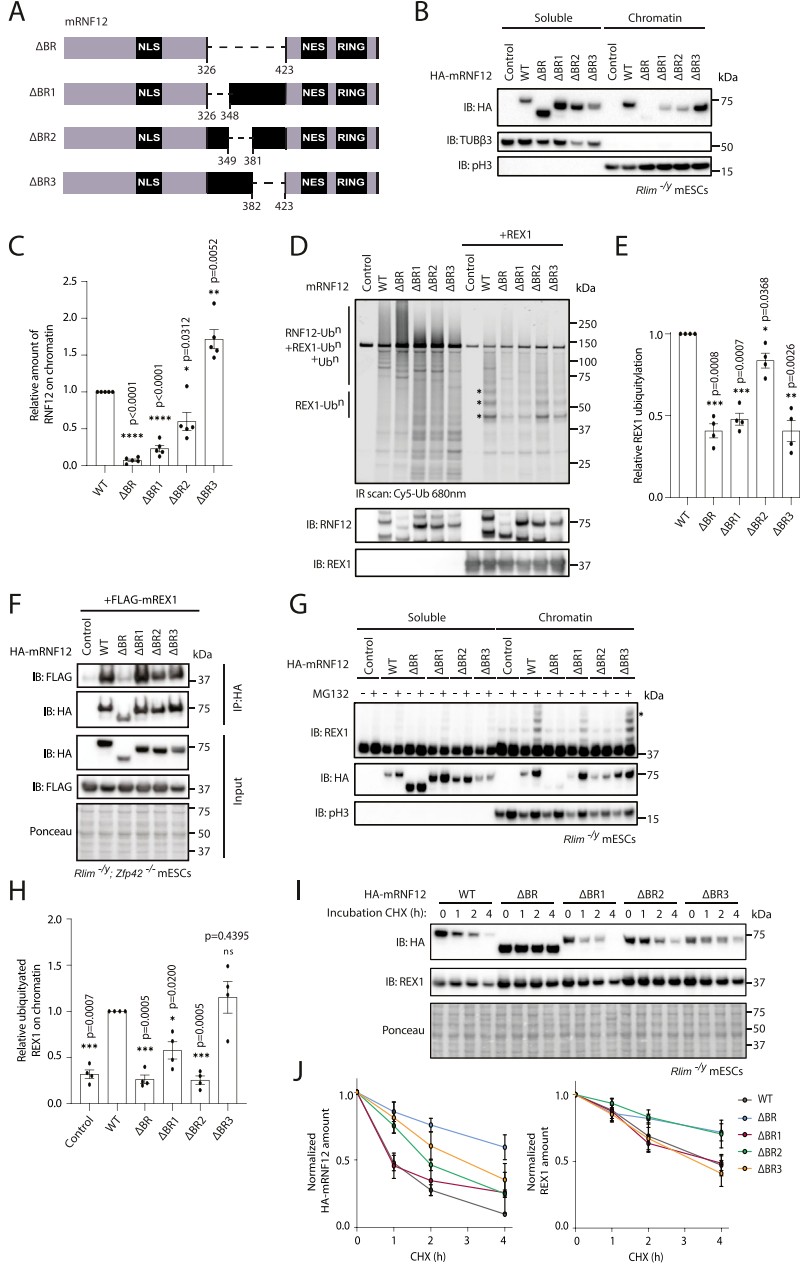

**Figure 5. RNF12 chromatin recruitment and substrate ubiquitylation are mediated by the basic region (BR).**
**(A)** Schematic representation of mouse RNF12 BR, BR1, BR2, and BR3 deletions. **(B)** RNF12 knock-out (*Rlim*[−/y]) mouse embryonic stem cells (mESCs) expressing either empty vector (control), HA-tagged mouse RNF12[WT], or RNF12 BR deletions were subjected to chromatin fractionation. HA-RNF12, REX1, βIII-tubulin (TUBβ3), and phospho-Ser10 Histone H3 (pH3) levels were determined by immunoblotting. Data are representative of n = 5 independent experiments. **(C)** Quantification of HA-RNF12 protein levels observed in the chromatin fraction in (B) expressed relative to RNF12[WT]. Data represented as mean ± SEM (n = 5). Statistical significance was determined by paired *t* test; two-sided, confidence level 95%. **(D)** In vitro REX1 substrate ubiquitylation assay of RNF12[WT] and RNF12 BR deletions. Top: fluorescently labelled ubiquitylated proteins were detected by 680 nm scan (Cy5-Ub). Ubiquitylated REX1 (REX1-Ub[n]) and RNF12 (RNF12-Ub[n]) signals are indicated. Bottom: immunoblot analysis of RNF12 (using anti-RNF12 mouse monoclonal antibody) and REX1 protein levels. Data are representative of n = 4 independent experiments. **(E)** REX1 ubiquitylation was quantified and normalized to total REX1. **(D)** The first three REX1 ubiquitylated bands (REX1-Ub[3]), indicated by asterisks in (D), were identified by comparison with control lacking REX1 substrate and quantified. Background correction was not applied because RNF12 preferentially ubiquitylates REX1, but in the absence of REX1 performs auto-ubiquitylation and/or forms free ubiquitin chains, creating variability in the control signal. Data represented as mean ± SEM (n = 4). Statistical significance was determined by paired *t* test; two-sided, confidence level 95%. **(F)** RNF12/REX1 double knock-out (*Rlim*[−/y]; *Zfp42*[−/−]) mESCs expressing FLAG-REX1 with either empty vector (control), HA-tagged mouse RNF12[WT], or the indicated HA-RNF12 deletion mutants were treated with MG132 for 2 h and HA-RNF12 immunoprecipitated. HA-RNF12 and FLAG-REX1 levels are determined by immunoblotting. Ponceau S staining is shown as a loading control. Data are representative of n = 4 independent experiments. **(G)** RNF12 knock-out (*Rlim*[−/y]) mESCs expressing either empty vector (control), HA-tagged mouse RNF12[WT], or HA-RNF12 BR deletions were treated with either DMSO (vehicle control) or MG132 for 1 h and subjected to chromatin fractionation. HA-RNF12, βIII-tubulin (TUBβ3), and phospho-Ser10 Histone H3 (pH3) levels were determined by immunoblotting. Data are representative of n = 4 independent experiments. **(H)** REX1 ubiquitylation in the chromatin fraction of MG132-treated mESCs from (G) was quantified by determining relative average intensity of the fourth ubiquitylated band (REX1-Ub[4]; indicated by an asterisk in (G) and normalizing to total REX1 levels. REX1 ubiquitylation is expressed relative to RNF12[WT]. Data represented as mean ± SEM (n = 4). Statistical significance was determined by paired *t* test; two-sided, confidence level 95%. **(I)** RNF12 knock-out (*Rlim*[−/y]) mESCs expressing HA-tagged mouse RNF12[WT] or HA-RNF12 BR deletions were treated with 350 µM cycloheximide (CHX) for the indicated times. HA-RNF12 and REX1 levels were determined by immunoblotting. Ponceau S staining is shown as a loading control. Data are representative of n = 4 independent experiments. **(J)** Quantification of data from (I) representing normalized HA-RNF12 and REX1 protein levels relative to control (0). Data represented as mean ± SEM (n = 4). Statistical significance of each deletion mutant compared with HA-RNF12[WT] was determined at 2 h for HA-RNF12 and at 4 h for REX1 by paired *t* test; two-sided, confidence level 95%. HA-RNF12: RNF12[ΔBR] (**) P = 0.0026, RNF12[ΔBR1] (ns) P = 0.6554, RNF12[ΔBR2] (*) P = 0.0393, and RNF12[ΔBR3] (*) P = 0.0186. REX1: RNF12[ΔBR] (*) P = 0.0322, RNF12[ΔBR1] (ns) P = 0.7590, RNF12[ΔBR2] (**) P = 0.0050, and RNF12[ΔBR3] (ns) P = 0.2781.
Source data are available for this figure.

in combination with chromatin fractionation, we measured REX1 ubiquitylation as previously. These data suggest that the RNF12 N-terminus is required for REX1 ubiquitylation (Fig 6D and E), in contrast with previous data indicating that the RNF12 N-terminus suppresses REX1 substrate recruitment (Fig 4F and G). To resolve these apparently contradictory results, we investigated the direct impact of the RNF12 N-terminal region on E3 ubiquitin ligase activity

in the presence of recombinant ubiquitin, UBE2D1 (E2) and UBE1 (E1) and REX1 in vitro. As shown previously (Fig 5D), RNF12[WT] specifically ubiquitylates REX1 substrate in vitro (Fig 6F and G). Deletion of the RNF12 N-terminal region significantly increases REX1 ubiquitylation (Fig 6F and G), suggesting that the RNF12 N-terminus acts to suppress chromatin recruitment and substrate engagement, and also to inhibit catalytic activity. Consistent with these impacts,

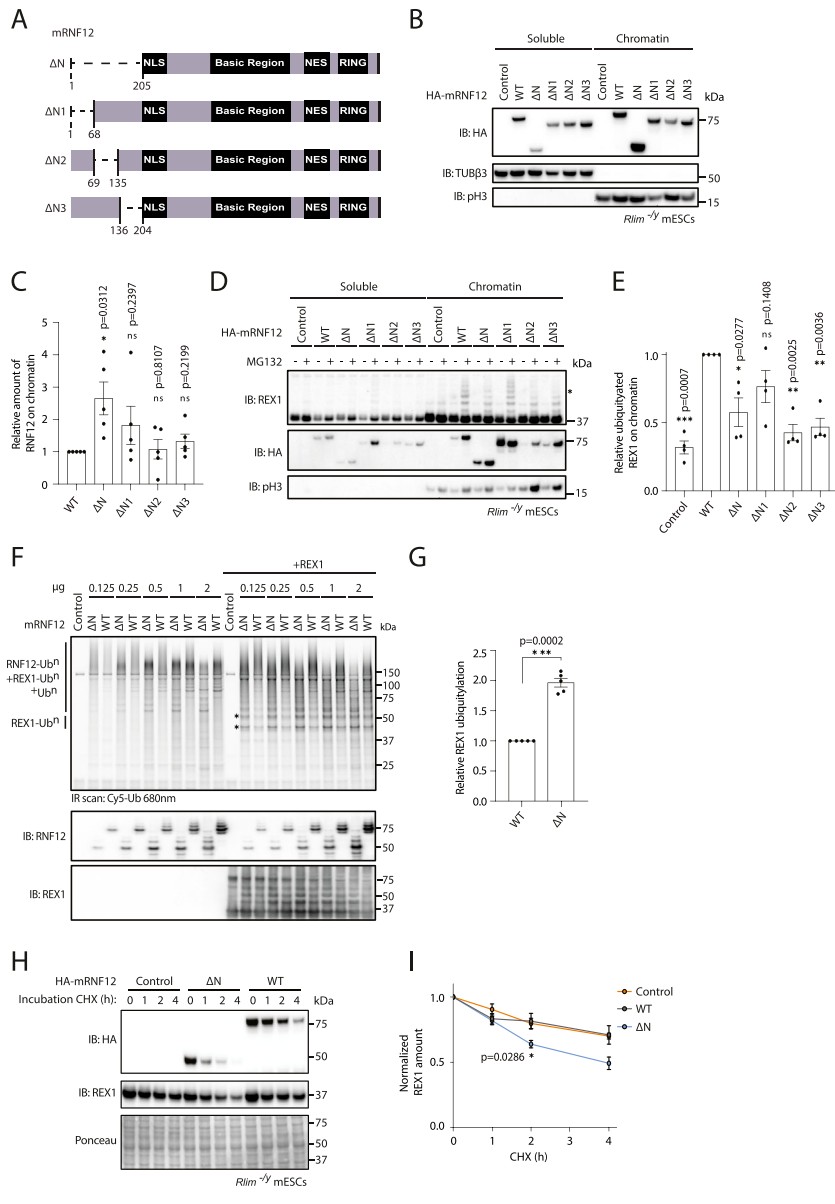

**Figure 6. RNF12/RLIM N-terminal region inhibits chromatin recruitment and substrate ubiquitylation.**
**(A)** Schematic representation of mouse RNF12 N-terminal deletions (ΔN, ΔN1, ΔN2, ΔN3). **(B)** RNF12 knock-out (*Rlim⁻/y*) mouse embryonic stem cells (mESCs) expressing HA-tagged mouse RNF12$^{WT}$ and N-terminal deletions were subjected to chromatin fractionation. HA-RNF12, βIII-tubulin (TUBβ3), and phospho-Ser10 Histone H3 (pH3) levels were determined by immunoblotting. Data are representative of n = 5 independent experiments. **(C)** Quantification of HA-RNF12 protein levels observed in the chromatin fraction in (B) expressed relative to RNF12$^{WT}$. Data represented as mean ± SEM (n = 5). Statistical significance was determined by paired *t* test; two-sided, confidence level 95%. **(D)** RNF12 knock-out (*Rlim⁻/y*) mESCs expressing HA-tagged mouse RNF12$^{WT}$ and N-terminal deletions were treated with MG132 for 1 h and subjected to chromatin fractionation. REX1, HA-RNF12, and phospho-Ser10 Histone H3 (pH3) levels were determined by immunoblotting. Data are representative of n = 4 independent experiments. **(E)** REX1 ubiquitylation in the chromatin fraction of MG132-treated mESCs from (D) was quantified by determining relative average intensity of the fourth ubiquitylated band (REX1-Ub⁴; indicated by an asterisk in (D)) and normalizing to total REX1 levels. REX1 ubiquitylation is expressed relative to RNF12$^{WT}$. Data represented as mean ± SEM (n = 4). Statistical significance was determined by paired *t* test; two-sided, confidence level 95%. **(F)** In vitro REX1 ubiquitylation assay containing increasing amounts of mRNF12$^{WT}$ and mRNF12$^{ΔN}$. Top: fluorescently labelled ubiquitylated proteins were detected by 680 nm scan (Cy5-Ub). Specific ubiquitylated REX1 (REX1-Ub$^n$) and RNF12 (RNF12-Ub$^n$) signals are indicated. Bottom: immunoblot analysis of RNF12 (using anti-RNF12 mouse monoclonal antibody) and REX1 protein levels. Data are representative of n = 5 independent experiments. **(G)** REX1 ubiquitylation was quantified and normalized to total REX1. The first two REX1 ubiquitylated bands (REX1-Ub²; indicated by asterisks in (F)) were identified by comparison with control lacking REX1 substrate and quantified. Background correction was not applied because RNF12 preferentially ubiquitylates REX1 but, in the absence of REX1, performs auto-ubiquitylation and/or forms free ubiquitin chains, creating variability in the control signal. Data represented as mean ± SEM (n = 4). Statistical significance was determined by paired *t* test; two-sided, confidence level 95%. **(H)** RNF12 knock-out (*Rlim⁻/y*) mESCs expressing either empty vector (control), HA-tagged mouse RNF12$^{WT}$, or RNF12$^{ΔN}$ were treated with 350 µM cycloheximide (CHX) for the indicated times. HA-RNF12 and REX1 levels were determined by immunoblotting. Ponceau S staining is shown as a loading control. Data are representative of n = 5 independent experiments.

**(I)** Quantification of normalized REX1 levels from (H) relative to control (0). Data represented as mean ± SEM (n = 5). Statistical significance of REX1 stability in *Rlim⁻/y* mESCs expressing HA-RNF12 N-terminal deletion compared with those expressing HA-RNF12$^{WT}$ was determined at 2 h by paired *t* test; two-sided, confidence level 95%. Source data are available for this figure.

RNF12 N-terminal deletion augments REX1 degradation in cells, under conditions where RNF12$^{WT}$ levels are limiting for REX1 processing (Fig 6H and I). Taken together, these data indicate an apparent autoinhibitory function of the RNF12 N-terminal region in suppressing chromatin recruitment, REX1 substrate recruitment, and ubiquitylation.

## RNF12 chromatin recruitment is required for transcription of the X-chromosome inactivation factor *Xist*

Finally, we sought to determine the functional importance of RNF12 chromatin recruitment for regulation of gene expression. One of the key functions of RNF12 during development is induction of imprinted X-chromosome inactivation (Shin et al, 2010; Barakat et al, 2011; Wang et al, 2016; Gontan et al, 2018), which occurs by relieving REX1-mediated transcription repression of the long-non-coding RNA *Xist* via REX1 ubiquitylation and degradation (Barakat et al, 2011; Gontan et al, 2012). Therefore, we used an assay for ectopic *Xist* induction by RNF12 expression in male mESCs (Jonkers et al, 2009; Shin et al, 2010; Barakat et al, 2011; Gontan et al, 2012, 2018), which serves as a sensitive readout of RNF12-dependent transcriptional responses mediated by REX1 degradation. As expected, *Xist* expression is low in male mESCs, but expression of HA-RNF12$^{WT}$ drives *Xist* transcription (Fig 7A and B). In contrast, expression of catalytically deficient RNF12$^{W576Y}$ fails to drive *Xist* induction (Fig 7A and B), indicating that *Xist* gene regulation is

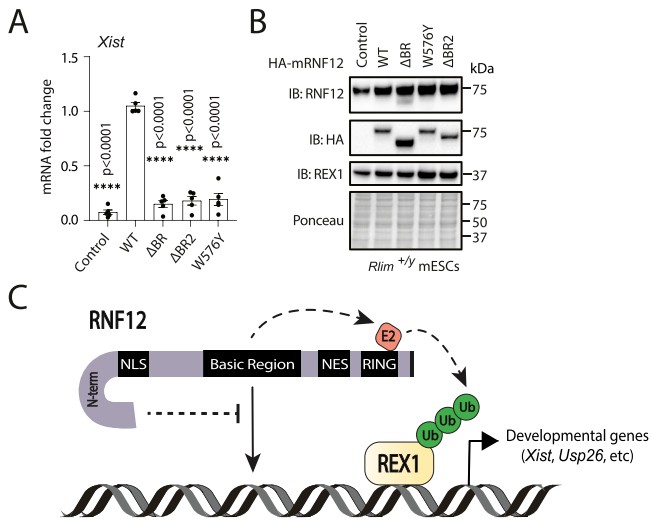

**Figure 7. RNF12 chromatin recruitment is required for target gene transcription.**
**(A)** *Rlim*$^{+/y}$ mouse embryonic stem cells (mESCs) expressing HA-tagged mouse RNF12$^{WT}$, RNF12$^{\Delta BR}$, RNF12$^{W576Y}$, and RNF12$^{\Delta BR2}$ and differentiated for 72 h. *Xist* RNA levels were normalized to *Gapdh* and represented as fold-change relative to RNF12$^{WT}$. Data represented as mean ± SEM (n = 5). Statistical significance was determined by paired *t* test; two-sided, confidence level 95%. **(B)** *Rlim*$^{+/y}$ mouse embryonic stem cells expressing HA-tagged mouse RNF12$^{WT}$, HA-RNF12$^{\Delta BR}$, HA-RNF12$^{W576Y}$, and HA-RNF12$^{\Delta BR2}$ were lysed, and total RNF12, HA-RNF12, and REX1 levels determined by immunoblotting. Ponceau staining is shown as a control. Data are representative of n = 5 independent experiments. **(C)** Model for how chromatin functions as an RNF12 regulatory platform. N-term = RNF12 N-terminal sequences. RNF12 recruitment to chromatin is mediated by the RNF12 BR, which is required for efficient REX1 ubiquitylation and regulation of RNF12-dependent genes. In an opposing manner, RNF12 N-terminal sequences supress chromatin recruitment and substrate ubiquitylation, conferring a previously unappreciated autoinhibitory mechanism. Note that the RNF12 BR is also involved in direct regulation of catalytic activity.
Source data are available for this figure.

dependent upon RNF12 E3 ubiquitin ligase activity. Similarly, expression of RNF12 deletions lacking either the entire BR or the BR2 region that significantly impacts chromatin recruitment but not substrate binding or catalysis, fails to drive *Xist* transcription (Fig 7A and B). These data therefore provide evidence that chromatin recruitment of RNF12 by the BR plays a key role developmental gene expression, as measured by *Xist* induction.

## Discussion

In this article, we uncover a critical role for chromatin in regulation of substrate ubiquitylation and downstream regulation of gene expression by the RING-type E3 ubiquitin ligase RNF12/RLIM. We show that RNF12 is recruited to specific DNA sequence motifs on chromatin, including co-recruitment at sites occupied by a key substrate, the transcription factor ZFP42/REX1. Recruitment to these specific locations facilitates RNF12-mediated ubiquitylation of REX1, thereby inducing expression of RNF12-REX1-dependent genes, as measured by the long-non-coding RNA *Xist*, which co-ordinates X-chromosome inactivation. Furthermore, we reveal the mechanism underpinning chromatin recruitment, whereby a

conserved RNF12 basic region (BR) independent of the catalytic RING domain is absolutely required. In addition, we show that the BR and another non-RING element at the N-terminal region perform key regulatory functions on chromatin. The RNF12 BR is required for chromatin recruitment, substrate engagement, and ubiquitylation, whereas the N-terminal region performs an auto-inhibitory function, which prevents chromatin recruitment, substrate engagement, and ubiquitylation (Fig 7C). In combination, this system is required for RNF12 substrate ubiquitylation and regulation of gene expression, providing insight into mechanisms by which ubiquitylation at gene promoters ensures specific transcriptional responses.

Yet to be resolved is structural detail of how RNF12 auto-inhibition, chromatin recruitment, substrate engagement, and ubiquitylation are coordinated. Although the RNF12 BR is required for chromatin recruitment, REX1 engagement and ubiquitylation, REX1 itself does not play a major role in RNF12 chromatin recruitment, suggesting that these mechanisms are separable. However, REX1 may be required to recruit RNF12 to specific genomic locations. Indeed, the RNF12 BR is required for interaction with both chromatin and REX1, which facilitates REX1 ubiquitylation on chromatin. Furthermore, the mechanism by which the N-terminal region inhibits chromatin recruitment and ubiquitylation is not yet known. Our data suggests that the N-terminus makes auto-inhibitory contacts to inhibit chromatin interaction and substrate engagement/ubiquitylation. In support of this notion, an RNF12 Alphafold structural prediction indicates that the N-terminal region may form direct contacts with the BR (Fig S3), which may in turn occlude chromatin and/or substrate interaction sites. Whether this negative regulatory system is released by chromatin recruitment or by another signal remains to be determined. Interestingly, there are reported phosphorylation sites in proximity to the RNF12 N-terminal region, which may modulate chromatin engagement and/or substrate ubiquitylation.

In this study, we also reveal that RNF12 is recruited to a genomic consensus motif, which potentially enables REX1 ubiquitylation at specific genomic locations. In that regard, we show that most REX1 ubiquitylation occurs on chromatin, presumably at sites of RNF12 co-recruitment. This prompts the exciting hypothesis that REX1 is specifically ubiquitylated by RNF12 at gene regulatory elements, which could in principle facilitate transcriptional processivity and dynamics. However, although REX1 appears to play a key role as an accessory factor, it is not required for global recruitment of RNF12 to chromatin, suggesting the existence of other factors that determine sites of chromatin engagement. Future work will explore whether transcriptional components other than REX1 play a role or whether RNF12 encodes capacity to directly engage chromatin via its specific DNA binding consensus.

Finally, as RNF12 is mutationally disrupted in patients with the X-linked intellectual disability disorder TOKAS, an attractive hypothesis states that RNF12 chromatin recruitment and the regulatory systems uncovered herein may be impacted by TOKAS variants. RNF12 TOKAS patient variants are found largely clustered in the catalytic RING domain or the BR. Therefore, a priority will be to determine the impact of RNF12 BR mutations on chromatin recruitment or whether these largely impact catalysis, as has been previously suggested (Bustos et al, 2018).

**Table 1.  Reagents summary table**

| Reagent or resource | Name | Reference or source | Identifiers | Additional information |
|---|---|---|---|---|
| Cell line (Mus Musculus, male) | *Rlim$^{+/y}$* mESCs | | | Parental male mouse Embryonic Stem Cell line from the laboratory of Janet Rossant, SickKids Research Institute, Toronto |
| Cell line (Mus Musculus, male) | *Rlim$^{-/y}$* mESCs | Bustos et al (2018) | | Male mouse Embryonic Stem Cell line from the Laboratory of Greg Findlay, MRC PPU, Dundee |
| Cell line (Mus Musculus, male) | *Rlim$^{-/y}$; Zfp42$^{-/-}$*mESCs | Bustos et al (2020) | | Male mouse Embryonic Stem Cell line from the Laboratory of Greg Findlay, MRC PPU, Dundee |
| Cell line (Mus Musculus, male) | RNF12 WT-KI mESCs | Bustos et al (2018) | | Male mouse Embryonic Stem Cell line from the Laboratory of Greg Findlay, MRC PPU, Dundee |
| Cell line (Mus Musculus, female) | *Rlim $^{\pm}$* mESCs | Jonkers et al (2009) | | WT female ESC lines F1 2-1 (129/Sv-Cast/Ei) |
| Cell line | HEK293T | ATCC | CRL-3216 | |
| Recombinant DNA reagent | 3xHA-TurboID-NLS_pcDNA3 | Addgene | #107171 | |
| Recombinant DNA reagent | pCW57.1 | Addgene | #41393 | |
| Recombinant DNA reagent | pCAG-2xFLAG-V5 vector | Gontan et al (2012) | | |
| Recombinant DNA reagent | pCAG-2xFLAG–V5-Rnf12 | Gontan et al (2012) | | |
| Recombinant DNA reagent | pCAG-2xFLAG–V5-*Rnf12*$^{H569A,C572A}$ | Gontan et al (2012) | | |
| Recombinant DNA reagent | pGEX-6P-1 | Cytiva | #28954648 | pGEX-6P-1 |
| Recombinant DNA reagent | Control; EV | MRC-PPU Reagents and Services | DU49023 | pCAGGS puro |
| Recombinant DNA reagent | mRNF12 WT | MRC-PPU Reagents and Services | DU53765 | pCAGGS puro mouse Rnf12 |
| Recombinant DNA reagent | mRNF12 | | | |
| W576Y | MRC-PPU Reagents and Services | DU50800 | pCAGGS puro mouse Rnf12 W576Y | |
| Recombinant DNA reagent | mRNF12 H569A,C572A | MRC-PPU Reagents and Services | DU53245 | pCAGGS puro mouse Rnf12 H569A C572A |
| Recombinant DNA reagent | HA-mRNF12 WT | MRC-PPU Reagents and Services | DU50854 | pCAGGS puro HA mouse Rnf12 |
| Recombinant DNA reagent | HA-mRNF12 ΔN | MRC-PPU Reagents and Services | DU53408 | pCAGGS puro HA mouse Rnf12 206–600 (end) |
| Recombinant DNA reagent | HA-mRNF12 ΔN1 | MRC-PPU Reagents and Services | DU73373 | pCAGGS puro HA mouse Rnf12 δ 1–68 |
| Recombinant DNA reagent | HA-mRNF12 ΔN2 | MRC-PPU Reagents and Services | DU73372 | pCAGGS puro HA mouse Rnf12 δ 69–135 |
| Recombinant DNA reagent | HA-mRNF12 ΔN3 | MRC-PPU Reagents and Services | DU73773 | pCAGGS puro HA mouse Rnf12 δ 136–204 |
| Recombinant DNA reagent | HA-mRNF12 ΔNLS | MRC-PPU Reagents and Services | DU53426 | pCAGGS puro HA mouse Rnf12 δ 206–226 |
| Recombinant DNA reagent | HA-mRNF12 ΔBR | MRC-PPU Reagents and Services | DU53422 | pCAGGS puro HA mouse Rnf12 δ 326–423 |

| Reagent or resource | Name | Reference or source | Identifiers | Additional information |
|---|---|---|---|---|
| Recombinant DNA reagent | HA-mRNF12 ΔBR1 | MRC-PPU Reagents and Services | DU73377 | pCAGGS puro HA mouse Rnf12 δ 326–348 |
| Recombinant DNA reagent | HA-mRNF12 ΔBR2 | MRC-PPU Reagents and Services | DU73376 | pCAGGS puro HA mouse Rnf12 δ 349–381 |
| Recombinant DNA reagent | HA-mRNF12 ΔBR3 | MRC-PPU Reagents and Services | DU73371 | pCAGGS puro HA mouse Rnf12 δ 382–423 |
| Recombinant DNA reagent | HA-mRNF12 ΔNES | MRC-PPU Reagents and Services | DU53405 | pCAGGS puro HA mouse Rnf12 δ 502–513 |
| Recombinant DNA reagent | HA-mRNF12 ΔRING | MRC-PPU Reagents and Services | DU53419 | pCAGGS puro HA mouse Rnf12 1–543 |
| Recombinant DNA reagent | HA-mRNF12 W576Y | MRC-PPU Reagents and Services | DU61086 | pCAGGS puro HA mouse Rnf12 W576Y |
| Recombinant DNA reagent | HA-mRNF12 Y326-A423 | MRC-PPU Reagents and Services | DU76275 | pCAGGS puro HA mouse Rnf12 Y326-A423 |
| Recombinant DNA reagent | HA-mRNF12 M1-R205 | MRC-PPU Reagents and Services | DU76276 | pCAGGS puro HA mouse Rnf12 M1-R205 |
| Recombinant DNA reagent | FLAG-mREX1 | MRC-PPU Reagents and Services | DU63525 | pCAGGS puro 3XFLAG mouse Rex1 |
| Recombinant DNA reagent | GST-mRNF12 ΔN | MRC-PPU Reagents and Services | DU73386 | pGEX6P mouse Rnf12 + NLS 206–600 |
| Recombinant DNA reagent | GST-mRNF12 WT | MRC-PPU Reagents and Services | DU49041 | pGEX6P1 mouse Rnf12 |
| Recombinant DNA reagent | GST-mRNF12 ΔBR | MRC-PPU Reagents and Services | DU73381 | pGEX6P1 mouse Rnf12 δ 326–423 |
| Recombinant DNA reagent | GST-mRNF12 ΔBR1 | MRC-PPU Reagents and Services | DU73380 | pGEX6P1 mouse Rnf12 δ 326–348 |
| Recombinant DNA reagent | GST-mRNF12 ΔBR2 | MRC-PPU Reagents and Services | DU73379 | pGEX6P1 mouse Rnf12 δ 349–381 |
| Recombinant DNA reagent | GST-mRNF12 ΔBR3 | MRC-PPU Reagents and Services | DU73378 | pGEX6P1 mouse Rnf12 δ 382–423 |
| Recombinant DNA reagent | 3xHA-TurboID | | | pCW57.1-3xHA-TurboID |
| Recombinant DNA reagent | 3xHA-TurboID-RNF12 | | | pCW57.1-3xHA-TurboID-RNF12 |
| Antibody | Anti-RNF12 (sheep polyclonal) | MRC-PPU Reagents and Services | Cat#S691D third bleed | WB: 1:1,000 |
| Antibody | Anti-RNF12 (mouse monoclonal) | Novus Biologicals | Cat#H00051132-M01 | WB: 1:1,000 |
| Antibody | Anti-REX1 (sheep polyclonal) | MRC-PPU Reagents and Services | Cat#DA136 fourth bleed | WB: 1:2,000 |
| Antibody | Anti-FLAG-HRP (mouse monoclonal) | Sigma-Aldrich | Cat#A8592 | WB: 1:10,000 |
| Antibody | Anti-HA-HRP (Rat monoclonal) | Roche | Cat#12013819001 | WB: 1:10,000 |
| Antibody | Anti-tubulin β 3 (mouse monoclonal) | BioLegend | Cat#801202 | WB: 1:1,000 |
| Antibody | Anti-pHistone H3 (S10) (rabbit polyclonal) | Cell Signalling Technology | Cat#9701S | WB: 1:1,000 |
| Antibody | Anti-HA (rabbit polyclonal) | Abcam | Cat#ab9110 | WB: 1:20,000 |
| Antibody | Anti-ERK1 (mouse polyclonal) | BD Biosciences | Cat#610408 | WB: 1:1,000 |
| Antibody | Anti-rabbit-HRP (rabbit polyclonal) | Invitrogen | Cat#G21234 | WB: 1:40,000 |

| Reagent or resource | Name | Reference or source | Identifiers | Additional information |
|---|---|---|---|---|
| Antibody | Anti-sheep-HRP (donkey polyclonal) | Thermo Fisher Scientific | Cat#16041 | WB: 1:10,000 |
| Antibody | Anti-rabbit-HRP (goat polyclonal) | Cell Signalling Technology | Cat#7074S | WB: 1:10,000 |
| Antibody | Anti-mouse-HRP (horse polyclonal) | Cell Signalling Technology | Cat#7076S | WB: 1:10,000 |
| Antibody | Biotin anti-HA (mouse monoclonal) | Covance | Cat#BIOT-101L | IF: 1:1,000 |
| Antibody | Anti-mouse AlexaFluor 568 nm (goat polyclonal) | Thermo Fisher Scientific | Cat#A-11004 | IF: 1:500 |
| Antibody | Anti-rabbit AlexaFluor 546 nm (goat polyclonal) | Thermo Fisher Scientific | Cat#A-11035 | IF: 1:500 |
| Antibody | AlexaFluor 488 nm-conjugated streptavidin | Thermo Fisher Scientific | Cat#S32354 | IF: 1:1,000 |
| PCR primers | Xist_F | This article | 213,742 | GGATCCTGCTTGAACTACTGC |
| PCR primers | Xist_R | This article | | CAGGCAATCCTTCTTCTTGAG |
| Peptide, recombinant protein | LIF | MRC-PPU Reagents and Services | DU1715 | GST-tagged LIF |
| Peptide, recombinant protein | UBE1 | MRC-PPU Reagents and Services | DU32888 | |
| Peptide, recombinant protein | UBE2D1 (UbcH5a) | MRC-PPU Reagents and Services | DU4315 | |
| Peptide, recombinant protein | REX1 | MRC-PPU Reagents and Services | DU53244 | |
| Chemical compound | MG132 | Sigma-Aldrich | Cat#474790 | Mroczkiewicz, Michał et al Journal of medicinal chemistry 53.4 (2010): 1,509–1,518. |
| Chemical compound | Cycloheximide | Sigma-Aldrich | Cat#C7698 | Schneider-Poetsch, Tilman et al Nature chemical biology 6.3 (2010): 209–217. |

# Materials and Methods

A full summary of reagents used in this study can be found in Table 1.

## mESC culture and transfection

Male mESCs were obtained from the laboratory of Janet Rossant (SickKids Research Institute, Toronto). RNF12 WT knock-in (WT-KI) (Bustos et al, 2018), RNF12 knock-out (*Rlim*$^{-/y}$) (Bustos et al, 2018) and RNF12/REX1 double knock-out (*Rlim*$^{-/y}$; *Zfp42*$^{-/-}$) (Bustos et al, 2020) mESCs were described previously. mESCs were cultured in 0.1% gelatin (wt/vol) coated plates in ES-DMEM containing 10% (vol/vol) FBS, 5% (vol/vol) knock-out serum replacement, 2 mM glutamine, 0.1 mM MEM, non-essential amino acids, penicillin/streptomycin, 1 mM sodium pyruvate (all Thermo Fisher Scientific), 0.1 mM $\beta$-mercaptoethanol (Sigma-Aldrich), and 20 ng/ml GST-tagged leukemia inhibitory factor (Medical Research Council Protein Phosphorylation and Ubiquitin Unit Reagents and Services [MRC-PPU R&S] http://mrcppureagents.dundee.ac.uk) at 37°C with 5% CO$_2$ in a water-saturated incubator. cDNA plasmid clones were transfected in mESCs with Lipofectamine LTX (Thermo Fisher Scientific) according to the manufacturer instructions. All cells were tested monthly for mycoplasma contamination.

## TurboID cell lines

TurboID stable mESC lines were generated using lentiviral transduction. HEK293T cells (CRL-3216; ATCC) were transfected with each construct and third-generation lentiviral packaging plasmids (VPK-206; Cell BioLabs) using Lipofectamine 3000 (Thermo Fisher Scientific) as per the manufacturer's recommendation. Transfected cells were incubated at 37°C for 6 h, replenished with fresh medium, and further incubated at 32°C for 72 h. The culture media was filtered through a 0.45 $\mu$m filter, concentrated by ultra-centrifugation (20,000$g$ and 4°C), resuspended in growth media, and added to mESCs along with Polybrene (4 $\mu$g/ml; Santa Cruz Biotechnology). 96 h after transduction, puromycin (6 $\mu$g/ml; Thermo Fisher Scientific) was added to select for transduced cells. Established cell lines were grown in 20 $\mu$g/ml puromycin. All cells were tested monthly for mycoplasma contamination.

## ChIP-seq cell lines

Cell lines stably expressing 2xFLAG-V5-RNF12 and 2xFLAG–V5-RNF12$^{H569A,C572A}$ were generated by electroporation of *Rlim*$^{+/-}$ (also termed *Rnf12*$^{+/-}$) female mESCs F1 2-1 (129/Sv-Cast/Ei) (Jonkers et al, 2009),

with pCAG-2xFLAG–V5-*Rnf12* or pCAG-2xFLAG–V5-*Rnf12*[H569A,C572A] vectors followed by puromycin selection. The coding sequence of *Rnf12* was amplified from mESC cDNA and cloned into a TOPO blunt vector (Invitrogen). *Rnf12*[H569A,C572A] mutant was generated by PCR-site-directed mutagenesis. For mammalian expression, the WT and mutant *Rnf12* coding sequences were subcloned into the pCAG-2xFLAG-V5 vector.

## cDNA plasmids

TurboID plasmids were made using In-Fusion Recombination (Takara Bio USA, Inc.). 3xHA-TurboID was amplified from 3xHA-TurboID-NLS pcDNA3 (plasmid #107171; Addgene) and inserted into empty pCW57.1 (plasmid #41393; Addgene) using the NheI and BamHI restriction enzyme (RE) sites, with the addition of AgeI RE site built into the 3' primer. 3xHA-TurboID pCW57.1 was used as the control plasmid. RNF12[WT] was amplified via PCR from pCAGGS RNF12 (MRC-PPU R&S) and inserted into 3xHA-TurboID pCW57.1 at AgeI and BamHI RE sites. All other cDNA plasmids are available from MRC-PPU R&S and were verified by DNA sequencing (MRC-PPU DNA Sequencing Service) using DYEnamic ET terminator chemistry (Amersham Biosciences) on Applied Biosystems 3730 automated capillary DNA sequencers.

## TurboID proximity-labelled protein purification

Large-scale TurboID pulldowns were performed in triplicate, as described in (May & Roux, 2019). In brief, three 15 cm plates per condition at $6 \times 10^4$ cells/cm$^2$ were plated in presence of doxycycline (1 mg/ml) (Thermo Fisher Scientific) for 18 h. Cells were then treated with 10 $\mu$M MG132 together with 50 $\mu$M biotin (Sigma-Aldrich) for 4 h to inhibit proteasome degradation and to induce biotinylation, respectively. Cells were washed twice with PBS and lysed in 8 M urea, 50 mM Tris pH 7.4 containing protease inhibitor (Thermo Fisher Scientific) and DTT, incubated with universal nuclease (Thermo Fisher Scientific), and sonicated to further shear DNA. Lysates were precleared with Gelatin Sepharose 4B beads (GE Healthcare) for 2 h and then incubated with Streptavidin Sepharose High Performance beads (GE Healthcare) for 4 h. Streptavidin beads were washed four times with 8 M urea, 50 mM Tris pH 7.4 wash buffer, and resuspended in 50 mM ammonium bicarbonate with 1 mM biotin. To analyse post-pulldown fractions by immunoblot, 10% of the post-pulldown bead fractions were used.

## Mass spectrometry analysis

Protein samples were reduced, alkylated, and digested on-bead using filter-aided sample preparation (Wiśniewski et al, 2009) with sequencing-grade modified porcine trypsin (Promega). Tryptic peptides were separated by reverse-phase XSelect CSH C18 2.5 $\mu$m resin (Waters) on an in-line 150 × 0.075 mm column using an UltiMate 3000 RSLCnano system (Thermo Fisher Scientific). Peptides were eluted using a 60-min gradient from 98:2 to 65:35 buffer A:B ratio (Buffer A = 0.1% formic acid, 0.5% acetonitrile, Buffer B = 0.1% formic acid, 99.9% acetonitrile). Eluted peptides were ionized by electrospray (2.4 kV), followed by mass spectrometric analysis on an Orbitrap Fusion Tribrid mass spectrometer (Thermo Fisher

Scientific). MS data were acquired using the FTMS analyzer in profile mode at a resolution of 240,000 over a range of 375–1,500 m/z. After HCD activation, MS/MS data were acquired using the ion trap analyzer in centroid mode and normal mass range with normalized collision energy of 28–31% depending on charge state and precursor selection range. Proteins were identified by database search using MaxQuant (Max Planck Institute) label-free quantification with a parent ion tolerance of 2.5 ppm and a fragment ion tolerance of 0.5 Da. Scaffold Q+S (Proteome Software) was used to verify MS/MS-based peptide and protein identifications. Protein identifications were accepted if they could be established with less than 1.0% false discovery and contained at least two identified peptides. Protein probabilities were assigned by the Protein Prophet algorithm (Nesvizhskii et al, 2003).

## ChIP-seq methodology

The ChIP-seq experiments were performed as described (Soler et al, 2011) with minor modifications. For the RNF12 ChIP-seq experiments, $1 \times 10^8$ undifferentiated female ESCs expressing V5-tagged RNF12, V5-tagged RNF12[H569A,C572A], and control WT ESCs were cultured without feeders until they reached 80% confluence. Cells were treated for 3 h with 15 $\mu$M MG132 proteasome inhibitor. All buffers used contained protease inhibitor cocktail tablet (Roche) and 15 $\mu$M MG132. The medium was removed, and cells were washed three times with PBS. Cells were then cross-linked by incubating with PBS containing 2 mM DSG (Thermo Fisher Scientific) for 45 min at room temperature (RT) on a rotating platform. After the incubation, cells were washed three times with PBS. In the last wash, formaldehyde was added to 1% final concentration and incubated for 10 min at RT, followed by the addition of glycine to a final concentration of 0.125 M, and cells were incubated for an additional 5 min at RT to quench the reaction. Cells were washed twice with ice-cold PBS, then scraped and collected in cold PBS. The fixed cell pellets were resuspended in lysis buffer (10 mM Tris–HCl pH 7.5, 1 mM EDTA, 0.5 mM EGTA) and incubated 10 min on ice. Samples were sonicated on ice using a Sanyo Soniprep 150 sonicator (amplitude 9, 37 cycles of 15 s on and 30 s off) to a DNA fragment size in the range of 300–800 nucleotides. The sonicated chromatin samples were centrifuged at 17,000$g$ for 5 min at 4°C. Chromatin was then diluted to a final volume of 10 ml with dilution buffer (0.01% SDS, 1.1% Triton X-100, 1.2 mM EDTA, 16.7 mM Tris–HCl pH 8.0, 167 mM NaCl), precleared, and immunoprecipitated overnight at 4°C with 60 $\mu$l of pre-blocked V5 agarose beads (Sigma-Aldrich) for each ChIP-seq experiment. Beads were washed twice with low salt buffer (0.1% SDS, 1% Triton X-100, 2 mM EDTA, 20 mM Tris–HCl pH 8.0, 150 mM NaCl), followed by two washes with high salt buffer (0.1% SDS, 1% Triton X-100, 2 mM EDTA, 20 mM Tris–HCl pH 8.0, 500 mM NaCl), two washes with LiCl buffer (0.25 M LiCl, 1% NP-40, 1% sodium deoxycholate, 1 mM EDTA, 10 mM Tris–HCl pH 8.0), and two washes with TE buffer (10 mM Tris–HCl pH 8.0, 1 mM EDTA). Each wash step was performed for 10 min at 4°C on a rotating platform. Chromatin was eluted with 500 $\mu$l of elution buffer (1% SDS; 0.1 M NaHCO$_3$ in H$_2$O). Chromatin was de-crosslinked by adding 20 $\mu$l of 5 M NaCl and incubating at 65°C for 4 h. Then, 10 $\mu$l of 0.5 M EDTA, 20 $\mu$l of 1 M Tris–HCl pH 6.5, 20 $\mu$g of proteinase K were added and incubated at 45°C for 1 h to degrade proteins. DNA was then Phenol–Chloroform

extracted and resuspended in 20 $\mu$l of H$_2$O. The concentration was then measured. Purified ChIP-DNA was prepared for sequencing according to the Illumina protocol and sequenced on a HiSeq 2000 sequencer (Illumina), resulting in 36-bp single reads.

## Pharmacological inhibition

Cycloheximide (CHX) was used at a final concentration of 350 $\mu$M and MG132 at a final concentration of 10 $\mu$M unless otherwise stated.

## mESC lysate preparation

mESCs were harvested using lysis buffer (20 mM Tris–HCl pH 7.4, 150 mM NaCl, 1 mM EDTA, 1% NP-40 [vol/vol], 0.5% sodium deoxycholate [wt/vol], 10 mM $\beta$-glycerophosphate, 10 mM sodium pyrophosphate, 1 mM NaF, 2 mM Na$_3$VO$_4$, and 0.1 U/ml Complete Protease Inhibitor Cocktail Tablets [Roche]). BCA Protein Assay Kit (Thermo Fisher Scientific) was used to measure protein concentration of lysates obtained according to manufacturer's instructions. A BSA protein curve was used as a standard to calculate protein concentration. For total protein extraction, lysis buffer was supplemented with 0.1% SDS, 2 mM MgCl$_2$, and Benzonase (1:500; Sigma-Aldrich).

## Chromatin fractionation

Method for separation of the soluble and chromatin fractions was based on Ballabeni et al (2004). mESCs were harvested by addition of Trypsin-EDTA (Gibco), transferred to a microcentrifuge tube, and washed with cold PBS. Cells were centrifuged at 500$g$ for 5 min and the resulting pellet was resuspended in CSK buffer (0.5% Triton X-100, 10 mM Hepes pH 7.4, 100 mM NaCl, 300 mM sucrose, 3 mM MgCl$_2$, 1 mM EGTA, 0.1 U/ml complete protease inhibitor cocktail tablets [Roche], and 20 $\mu$l/ml of 50X phosphatase inhibitor cocktail [5 mM NaF, 1 mM Na$_3$VO$_4$, 1 mM sodium pyrophosphate, 1 mM $\beta$-glycerophosphate]). Samples were incubated on ice for 5 min, and after centrifugation at 1,350$g$ for 5 min, the supernatant (soluble fraction) was saved to a new microcentrifuge tube. Pellet was washed three times with CSK buffer (centrifugations were for 3 min at 1,350$g$) and the final pellet was resuspended with NaCl buffer (0.1% Triton X-100, 50 mM Tris–HCL pH 7.4, 250 mM NaCl, 1 mM EDTA, 50 mM NaF, protease and phosphatase inhibitors, 2 mM MgCl$_2$ benzonase [1:500; Sigma-Aldrich]). Samples were incubated in NaCl buffer on ice for 30 min with resuspension every 10 min. Samples were then centrifuged at 16,000$g$ for 15 min, and the supernatant (chromatin fraction) was saved. The chromatin fraction was then centrifuged at 16,000$g$ for 15 min to remove any chromatin contamination, and supernatant was used for further analysis.

## Immunoprecipitation

For HA-tagged protein immunoprecipitation, 10 $\mu$l Pierce Anti-HA Magnetic Beads (Thermo Fisher Scientific) were used. Beads were washed three times with lysis buffer and incubated with 1 mg mESC protein lysate overnight at 4°C. Beads were then washed three times with lysis buffer supplemented with 500 mM NaCl. In each step, beads were separated using a magnetic stand, and the supernatant was discarded. LDS sample buffer was used to elute proteins bound to beads, and samples were heated for 5 min at 95°C.

## Immunoblotting

Commercial NuPAGE 4–12% Bis–Tris SDS–PAGE gels (Thermo Fisher Scientific) were used to load denatured protein samples or protein eluates from pulldown experiments. SDS–PAGE gels were then transferred to polyvinylidene fluoride membranes (Merck Millipore) and incubated with primary antibodies diluted in TBS-T (20 mM Tris–HCl pH 7.5, 150 mM NaCl supplemented with 0.2% [vol/vol] Tween-20 [Sigma-Aldrich]) containing 5% non-fat milk buffer (wt/vol) at 4°C overnight. FLAG-HRP and HA-HRP conjugated primaries antibodies were incubated for 1 h at RT. Membranes were then washed three times with TBS-T and incubated with secondary antibody for 1 h at RT. Finally, membranes were washed three times with TBS-T and subjected to chemiluminescence detection with Immobilon Western Chemiluminescent HRP substrate (Merck Millipore) using a Gel-Doc XR+ System (Bio-Rad). Images were analysed and quantified using Image Lab software (Bio-Rad).

## Protein purification

Mouse RNF12$^{WT}$, RNF12$^{W576Y}$ and RNF12$^{\Delta N}$, RNF12$^{\Delta BR}$, RNF12$^{\Delta BR1}$, RNF12$^{\Delta BR2}$, and RNF12$^{\Delta BR3}$ mutants were cloned into pGEX-6P-1 (Cytiva). GST-tagged proteins were purified from BL21-CodonPlus (DE3)-RIPL Competent *E. coli* (Agilent, 230280) as follows; colonies from a LB ampicillin (100 $\mu$g/ml) plate were transferred into liquid LB media supplemented with ampicillin (1:1,000 dilution) and cultured in a 2 litre flask at 37°C until OD600 reached 0.8. 10 $\mu$M IPTG (Sigma-Aldrich) was added to induce protein expression, and bacteria were incubated at 15°C with shaking at 180 rpm overnight. Bacteria were then harvested at 4,200$g$ in a JS 4.2 series rotor (Beckman Coulter) for 30 min at 4°C, and the resulting pellet was resuspended in 40 ml of lysis buffer (50 mM Tris–HCl pH 7.5, 150 mM NaCl, 10% glycerol, 1 mM DTT, and two tablets of Complete Protease Inhibitor Cocktail Tablets [Roche] per 100 ml lysis buffer). Bacteria were lysed by 2 min sonication with 15 s pulses on/off, and the extract was centrifuged at 40,000$g$ for 25 min at 4°C. The supernatant was then incubated with Glutathione Sepharose 4B beads (MRC-PPU R&S) for 90 min on a rotating wheel at 4°C. Samples were then washed three times with protein buffer (50 mM Tris–HCl pH 7.5, 150 mM NaCl, 10% glycerol, 1 mM DTT), and proteins were cleaved from GST and eluted from beads using PreScission Protease (MRC-PPU R&S) at 4°C overnight. Supernatant was separated from beads using a Poly-Prep Chromatography Column (Bio-Rad) and concentrated using an Amicon Ultra-15 Centrifugal Filter Unit 10 kD molecular weight cut-off (Millipore). Protein samples were aliquoted and flash-frozen in liquid nitrogen for storage at –80°C. Recombinant ACHE (acetylcholinesterase) protein was produced by Florent Colomb in Dr. Henry McSorley's laboratory (School of Life Sciences, University of Dundee) as described previously (Vacca et al, 2020).

## RNF12 in vitro ubiquitylation assays

RNF12 recombinant protein (140 nM) was incubated with 20 µl ubiquitylation mix containing 0.1 µM UBE1, 0.05 µM UBE2D1 (UBCH5A), 1.5 µg REX1, 2 µM Cy5-Ubiquitin (South Bay Bio), 0.5 mM Tris (2-carboxyethyl) phosphine (TCEP) pH 7.5, 5 mM ATP, 50 mM Tris–HCl pH 7.5 and 5 mM $MgCl_2$. Reactions were incubated for 30 min at 30°C, stopped with 2x LDS-reducing agent mix and heated for 5 min at 95°C. REX1 recombinant protein and UBE1 and UBE2D1 enzymes were produced by MRC-PPU R&S and purified via standard protocols (http://mrcppureagents.dundee.ac.uk/).

## EMSA

pGEX6P-1 plasmid DNA was linearized by BamHI and NotI restriction enzymes and further purified using NucleoSpin Gel and PCR Clean-up Kit (Thermo Fisher Scientific) according to the manufacturer's instructions. 0.05 µg of linearized plasmid was incubated with 0.2–0.8 µg of recombinant proteins in 10 µl TE buffer (10 mM Tris–HCl pH 8.0, 1 mM EDTA) at 37°C for 1 h. Samples were run on a 0.8% agarose gel and analysed using a Chemidoc Imaging System (Bio-Rad).

## Extraction of RNA and quantitative RT–PCR

mESCs transfected with the indicated cDNA plasmids were cultured for 48 h until confluent. For *Xist* induction analysis, *Rlim$^{+/y}$* mESCs were transfected as described and cultured for 72 h in LIF-deficient media before lysis. RNA was extracted using an Omega total RNA extraction kit (Omega Biotek) (column-based system) according to the manufacturer's instructions. The obtained RNA was then converted to cDNA using the iScript cDNA synthesis kit (Bio-Rad) according to the manufacturer's instructions. qRT-PCR primers (Life Technologies) were 20–24 bp with a melting temperature of 58–62°C. Sequences were either acquired from PrimerBank database (https://pga.mgh.harvard.edu/primerbank/) or designed with the use of the Primer3 software. Specificity of each primer was predicted in silico with the use of the NCBI Primer-Blast software (https://www.ncbi.nlm.nih.gov/tools/primer-blast/). qRT-PCR was performed using a SsoFast EvaGreen Supermix (Bio-Rad) in 384-well plates and a CFX384 real-time PCR system (Bio-Rad). Each sample consisted of 10 µl of a master mix containing 5.5 µl of SYBR Green, 440 nM forward and reverse primers, 1 µl cDNA, and nuclease-free water. Relative RNA levels were expressed using the $\Delta\Delta C_t$ method and normalized to *Gapdh* expression. Data were analysed in Excel software and plotted making use of GraphPad Prism 9.3.0 software.

## Immunofluorescence

Cells grown on gelatin-coated glass coverslips were fixed in 3% (wt/vol) PFA/PBS for 10 min and permeabilized by 0.4% (wt/vol) Triton X-100/PBS for 15 min. Cells were then blocked with a 1% fish gelatin (wt/vol) in PBS solution for 30 min in a humid chamber. Primary antibodies were diluted in 1% fish gelatin (wt/vol) in PBS solution and added to cells for 2 h at RT in a humid chamber. Cells were washed three times with PBS, and secondary antibodies were tagged to a fluorophore diluted in 1% fish gelatin (wt/vol) in PBS.

For labelling RNF12 deletion mutants, rabbit anti-hemagglutinin (HA) was used (Abcam). The primary antibody was detected using Alexa Fluor 546–conjugated goat anti-rabbit (Thermo Fisher Scientific). For labelling 3xHA TurboID fusion proteins, a mouse anti-HA antibody was used (Covance). The primary antibody was detected using Alexa Fluor 568–conjugated goat anti-mouse. Alexa Fluor 488–conjugated streptavidin (Thermo Fisher Scientific) was used to detect biotinylated proteins. DNA was detected with Hoechst dye 33342 (Thermo Fisher Scientific). Coverslips were mounted using 10% (wt/vol) Mowiol 4-88 (Polysciences). Epifluorescence images were captured as z-projections using a Nikon A1R confocal microscope and analysed by the NIS-Elements software. For localization of RNF12 deletion mutants, images were acquired using a Leica-SP8 confocal laser scanning microscope (63x oil immersion objective, NA 1.4) and processed using FlowJo.

## ChIP-seq data analysis

The SNPs in the 129/Sv and Cast/Ei lines were downloaded from the Sanger Institute (v.5 SNP142) (Keane et al, 2011). These were used as input for SNPsplit v0.3.4 (Krueger & Andrews, 2016) to construct an N-masked reference genome based on mm10 in which all SNPs between 129/Sv and Cast/Ei are masked. The ChIP-seq reads were trimmed and aligned to the N-masked reference genome using Trim Galore v0.6.7 (Krueger, 2015) and Bowtie2 v2.5.0 (Kim et al, 2015), respectively. SNPsplit was then used to assign the reads to either the 129/Sv or Cast/Ei bam file based on the best alignment or to an unassigned bam file if mapping to a region without allele-specific SNPs. The allele-specific and unassigned bam files were merged into a composite bam file using Samtools v1.10 (Li et al, 2009).

Peaks were called from the merged bam files using macs2 v2.2.7.1 (Feng et al, 2012) callpeak with narrow and default settings. For visualization, the tracks were normalized using macs2 bdgcmp with the Poisson *P*-value as normalization method. Peaks from the different transcription factors were compared using ChIPseeker v1.34.0 (Yu et al, 2015). We plotted the peak annotation using the functions annotatePeak and plotAnnoBar. Enrichment at the transcription start sites was visualized using plotAvgProf. For each transcription factor, we searched for overlapping motifs by running bedtools v2.30.0 (Quinlan & Hall, 2010), getfasta to get the sequences of the peaks, and meme-chip v5.5.2 (Machanick & Bailey, 2011) from the meme-3 suite (Bailey et al, 2015) using the JASPAR 2018 motif database (Khan et al, 2018).

The correlation between the datasets was analysed using deeptools v3.5.1 (Ramírez et al, 2016) multiBigwigSummary bins with 1,000 bp bins and the input-normalized bigwigs and plotted using deeptools plotCorrelation (--corMethod pearson --removeOutliers --log1p). Intervene v0.6.5 (Khan & Mathelier, 2017) was used to evaluate the genomic overlap between significant peaks. As the overlap highly depends on the thresholds used during peak calling and differences in background level, we also quantitatively compared the ChIP-seq datasets using MAnorm v1.3.0 (Shao et al, 2012) with default settings, resulting in lists of common and unique peaks. Heatmaps showing the enrichment of the input-normalized tracks at common and shared peaks were plotted using deeptools

computeMatrix (--referencePoint center --averageTypeBins means -b 3,000 -a 3,000 -bs 50) and plotHeatmap. We compared the motifs present at unique peaks by running homer v4.11 (Heinz et al, 2010) findMotifsGenome.pl with default settings on each subset of peaks. The motif results of the unique peaks were visualized in a scatter plot comparing the *P*-values of each motif.

### Protein sequence analysis

RNF12 protein alignment was performed using Clustal Omega (EMBL-EBI), and graphical representation was generated using ESPript (Robert & Gouet, 2014).

### Data analysis

Data are presented as mean ± SEM of biological replicates, where individual points represent a single biological replicate. Statistical significance was determined by means of paired *t* test. GraphPad Prism V9.00 software was used for representation purposes, and differences were statistically significant when $P < 0.05$. Immunoblots were quantified using the densitometric analysis in Image Lab software, and data are presented as mean ± SEM of at least three biological replicates. In qRT-PCR experiments, two technical replicates were included per sample and were shown as an average for quantification. Immunofluorescence images were analysed using Fiji (ImageJ) software.

Downstream analysis of TurboID data was performed using Perseus (version 2.0.10.0) and Curtain 2.0 (https://curtain.proteo.info/#/ developed by Toan Phung, Dario Alessi laboratory). The MaxQuant output was loaded into Perseus and the data matrix filtered to remove proteins only identified by site, reverse proteins, and potential contaminants. Label-free quantification values were transformed by $\log_2$, then data were filtered for valid values (at least two valid values in one group). A two-sample *t* test was performed and *P*-values adjusted using the Benjamini-Hochberg multiple hypothesis correction. Data were then exported for inputting into Curtain to generate a volcano plot and proteins with a >twofold increase were studied. Identification of genes annotated with nuclear localisation and/or function, and gene set enrichment analysis, were performed using the DAVID bioinformatics tool.

## Data Availability

The RNF12 ChIP-seq data generated in this study have been submitted to the NCBI Gene Expression Omnibus (GEO) under accession number GSE236354. Raw mass-spectrometry data used in this study has been deposited to the ProteomeXchange Consortium via the PRIDE (Perez-Riverol et al, 2022) partner repository (www.ebi.ac.uk/pride) with the following dataset identifiers; RNF12 TurboID mass-spectrometry (PXD046733) and RNF12 affinity purification mass-spectrometry data from (Gontan et al, 2012) (PXD047197).

## Supplementary Information

## Acknowledgements

C Espejo-Serrano was funded by a Wellcome Trust 4-yr PhD studentship and a core grant to the MRC-PPU (MC_UU_12016). C Aitken was funded by a core grant to the MRC-PPU (MC_UU_12016). GM Findlay was funded by a Wellcome Trust/Royal Society Sir Henry Dale Fellowship (211209/Z/18/Z). BF Tan, J Gribnau, and C Gontan were funded by a VIDI grant (09150172110079) from the Dutch Research Council (NWO). The Biochemistry Core at Sanford Research and the IDeA National Resource for Quantitative Proteomics at UAMS are funded by the National Institutes of Health (grants P20 GM103620 and R24 GM137786, respectively). The histology and imaging core at Sanford Research are funded by the National Institutes of Health (grant P30 GM145398). We thank Toan Phung (laboratory of Prof. Dario Alessi, MRC-PPU) and Frederic Lamoliatte (MRC-PPU) for help with CURTAIN data analysis, Dr Renata Soares (MRC-PPU) for mass-spectrometry data handling, Dr Rachel Toth and Dr Thomas Macartney (MRC-PPU R&S) for cloning, Dr Axel Knebel and Dr James Hastie (MRC-PPU R&S) for protein purification, and Dr Florent Colomb and Dr Henry McSorley (School of Life Sciences, University of Dundee) for help with EMSA assays and recombinant ACHE protein.

### Author Contributions

C Espejo-Serrano: conceptualization, data curation, formal analysis, validation, investigation, visualization, methodology, and writing—original draft, review, and editing.
C Aitken: data curation, formal analysis, validation, investigation, and visualization.
BF Tan: data curation, formal analysis, investigation, and visualization.
DG May: data curation, formal analysis, and investigation.
RJ Chrisopulos: data curation, formal analysis, and investigation.
KJ Roux: data curation, formal analysis, and investigation.
JAA Demmers: data curation, formal analysis, and investigation.
SG Mackintosh: conceptualization, data curation, formal analysis, supervision, funding acquisition, investigation, and project administration.
J Gribnau: conceptualization, data curation, formal analysis, supervision, funding acquisition, validation, investigation, visualization, project administration, and writing—review and editing.
F Bustos: conceptualization, resources, data curation, formal analysis, supervision, funding acquisition, validation, investigation, visualization, project administration, and writing—review and editing.
C Gontan: conceptualization, resources, data curation, formal analysis, supervision, funding acquisition, validation, investigation, visualization, project administration, and writing—original draft, review, and editing.
GM Findlay: conceptualization, resources, data curation, formal analysis, supervision, funding acquisition, visualization, project administration, and writing—original draft, review, and editing.

**Conflict of Interest Statement**

The authors declare that they have no conflict of interest.

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
