## [Reviewer comments · Life Science Alliance]

Life Science Alliance

Chromatin targeting of the RNF12/RLIM E3 ubiquitin ligase controls transcriptional responses

Carmen Espejo-Serrano, Catriona Aitken, Beatrice Tan, Danielle May, Rachel Chrisopulos, Kyle Roux, Jeroen Demmers, Samuel Mackintosh, Joost Gribnau, Francisco Bustos, Cristina Gontan, and Greg Findlay

DOI: <https://doi.org/10.26508/lsa.202302282>

Corresponding author(s): Greg Findlay, MRC Protein Phosphorylation and Ubiquitylation Unit

Review Timeline:	Submission Date:	2023-07-18
	Editorial Decision:	2023-08-07
	Revision Received:	2023-11-28
	Editorial Decision:	2023-12-18
	Revision Received:	2023-12-21
	Accepted:	2023-12-22

Transaction Report:

August 7, 2023

Re: Life Science Alliance manuscript #LSA-2023-02282

Dr. Greg Findlay
MRC Protein Phosphorylation and Ubiquitylation Unit
School of Life Sciences
Australia

Dear Dr. Findlay,

Thank you for submitting your manuscript entitled "Genomic targeting of the RNF12/RLIM E3 ligase selectively programmes developmental transcription" to Life Science Alliance. The manuscript was assessed by expert reviewers, whose comments are appended to this letter. We invite you to submit a revised manuscript addressing the Reviewer comments.

Thank you for this interesting contribution to Life Science Alliance. We are looking forward to receiving your revised manuscript.

Sincerely,

B. MANUSCRIPT ORGANIZATION AND FORMATTING:

Reviewer #1 (Comments to the Authors (Required)):

The manuscript by Daudén et al focuses on deciphering how the E3 ligase activity of RNF12, a RING E3 ligase, is regulated. Initial experiments focus on identifying RNF12 partner proteins. The initial data obtained leads them to focus on the nuclear fraction as a key substrate of RNF12 is Rex1, which is a transcription factor with an important role in development. Although they take an untargeted approach their focus is on the known substrate protein Rex1, which the Findlay group have analysed previously (e.g. Bustos et al 2018). Using various assays they identify details of chromatin recruitment of RNF12, including a preferred binding site.

Following establishment of a cellular assay that enables them to assess Rex ubiquitylation in cells the authors utilise a number of deletion constructs to probe to function of different parts of RNF12. Based on these studies they conclude:

- The N-terminal 200 residues inhibit chromatin recruitment
- The basic residues are important for chromatin recruitment & Rex1 ubiquitination
- These two regions therefore have opposing effects on chromatin recruitment, substrate binding and ubiquitination.

The manuscript is relatively well written and will be of interest to those interested in the function of RNF12. However, my major concern is the reliance on deletion studies. While additional studies would be ideal, the reader should be made aware of this potentially significant caveat.

Main points:

- 1) In figure 2 the authors use the RNF12 H569A/C572A mutant as a control. This mutant will result in unfolding of the RING domain which will likely influence partner protein binding. The RNF12-W576Y mutant may have been a better control. This should be commented on.
- 2) Deletion studies are used throughout the manuscript to attribute functions to different parts of RNF12. While these studies provide clues about function, significant caveats remain as deletion studies of this type end up bringing distant sequences into close proximity (i.e. adjacent!) and this can create novel functions. This is a significant concern and it would be helpful if the deletion studies were supported by other data i.e. point mutations or assessment of the proposed domains i.e. does the Basic domain alone bind to chromatin?
- 3) In a number of the experiments/figures the authors rely on fractionation of proteins into nuclear and soluble fractions. It would be helpful to show in the supplementary material the total protein expression as well as the protein in the nuclear/soluble fractions. This would help give the author a better idea of protein levels and ascertain if any of the deletion proteins cause protein aggregation/mislocalised.
- 4) Quantification of Rex1 ubiquitination in Figure 5d looks challenging. The details of how this was done should be outlined in the figure legend. For example it would be helpful to know how the background intensity in the -Rex1 samples was treated as the -Rex1 samples for the different RNF12 proteins vary quite a bit. It may be better to have probed Rex1 as done in Fig 3a/6d. This figure needs further explanation.

Minor points:

- 1) It would be better if panel 4c was below/adjacent to panel 4b.
- 2) Page 21, line 5: In the current format ref 35 should be referred to by first author.
- 3) It would be helpful to include the sequence of RNF12 - at least the basic region which they state is conserved, but don't show, in the supp material.
- 4) Figure 4f/g is not intuitive and the authors may wish to consider how to improve clarity.
- 5) In the discussion the authors refer to the alphafold model and interactions. It would be good to include this as a supplementary figure.

Reviewer #2 (Comments to the Authors (Required)):

In this study, the authors explore molecular underpinnings of the function of the E3 ubiquitin ligase RNF12/RLIM. TurboID

analysis to identify proximal interactors of RNF12 seems to unravel a high proportion of chromatin-associated factors, suggesting that RNF12's functions are associated with chromatin, which the authors went on to explore. ChIP-seq of RNF12 revealed discrete binding across the genome, most but not all overlapping with its main target, the transcription factor REX1. Consistently, the authors claim that REX1 is specifically ubiquitinated on chromatin, which seems to be the case but this part needs to be clarified (see below). The authors also showed that RNF12 associates with chromatin in the absence of REX1. Moreover, the authors functionally dissected different protein domains of RNF12 to identify which ones are responsible for the association to chromatin and/or ubiquitylation functions - the emerging picture is that RNF12 associates with chromatin via a domain rich in basic residues, and the chromatin recruitment is important for ubiquitylation of REX1, while N-terminal sequences have a suppressing effect on chromatin recruitment and subsequent substrate ubiquitylation. Finally, the authors show the importance of the basic residue region for RNF12 function by analysing expression of Xist, an RNF12-REX1 target, in XY mESC using an overexpression assay - while overexpression of wildtype RNF12 leads to Xist upregulation, this is not the case for RNF12 versions without the basic region.

This is an interesting study, but there seems to be overinterpretation of results and the manuscript could also be improved for clarity and accuracy; for instance, as pointed out below, several statements in the Discussion do not seem to be backed up by the evidence shown (or the authors do not make their point come across clearly).

- The title could be improved, the current form is very cryptic, i.e., difficult to understand what it means / what was done and discovered. Moreover, "programmes developmental transcription" sounds too big when the authors checked only one gene (see more comments below). Still in the title and across the paper, the expression "genomic targeting" can be misleading - my first thought was that the authors meant they used a knockout approach, when I suppose they mean chromatin association/recruitment? The strongest point of the paper is the "novelty" of the RNF12 E3 ubiquitin ligase associating with chromatin and I think the title could reflect just that - as the authors mention in the discussion, "we uncover a critical role for chromatin in regulation of substrate ubiquitylation"
- Please explain what MG132 is the first time it appears, and the rationale to use it - for the non-specialists it is not obvious.
- In the introduction, the authors state that "RNF12 regulates highly specific gene expression programmes involved in X-chromosome inactivation, neurodevelopment and gametogenesis" - what about the functions in the mammary gland? (at least one example in PMID: 22541433)
- In Fig. 1B, the nuclear localisation of HA-TurboID-RNF12 is compared to HA-TurboID but should be compared to RNF12?
- The authors show that there is little overlap between their TurboID approach and the RNF12 affinity-purification mass-spec approach done previously by Gontan et al 2012, and state that this suggests that "the TurboID approach is able to reveal previously undiscovered RNF12 proximal proteins". Could it suggest instead that the approach picks up a lot of "noise"? How can these two possibilities be distinguished?
- Related to the previous point, the authors consider results of the TurboID approach only based on fold-change (higher than 2) and dismissing statistical significance. What is the rationale for this? Shouldn't the authors only consider hits for which the p-value is below at least 0.05? How does this affect the conclusion, i.e., that there is a high proportion of factors associated to chromatin?
- Could the authors add a line explaining the choice of DAVID (instead of GO or GSEA, for instance)?
- Still regarding Fig. 1D and the DAVID gene set enrichment analysis: RNF12 proximity labelled proteins are significantly enriched for chromatin-dependent functions - compared to...? What is the subset of proteins being compared to? Given that the authors already pre-selected proteins with annotated nuclear localisation and/or function, was the "control group" pre-selected for this as well? Otherwise, it is very likely that a group of proteins selected for nuclear localisation and/or function will be enriched in chromatin-related functions when compared to all proteins?
- Could the authors comment on the fact that more than half of the sites bound by RNF12-WT and RNF12-H569A,C572A do not coincide? Isn't this unexpected?
- Given the central message of the paper being RNF12 association to chromatin, I think it would be worth exploring further the RNF12 "binding" sites. For sites where there is no Rex1, are there other factors enriched? (based on published datasets and/or motif analysis?) How would this compare to other RNF12 targets?
- The authors report a "strong enrichment of RNF12 and REX1 at transcriptional start sites" - are these the same TSS for both RNF12 and REX1? What is the overlap?
- Could the authors add in the Introduction and/or Discussion what is known - or not known - about how RNF12 and Rex1 physically interact, if at all?

- The Results section about Fig. 3 is difficult to understand for lack of explanations and details. The authors say they measured "ubiquitylation of REX1" (both in the Results sections and Figure legends) but how was this done? Did they use an antibody specific for ubiquitylated REX1? I don't think this exists? And there is no mention of such in the Methods. So I assume that in Fig. 3B the authors are simply quantifying REX1 signal (and not ubiquitylated REX1), except that it's in the presence of MG132?
- Moreover, for a proper assessment of amount of ubiquitylated REX1 in the chromatin fraction compared to soluble, and to conclude that there is a chromatin-specific effect, I think the results from Fig. 3A have to be compared to results without MG132 treatment (maybe results from Fig. 2C could be used?)? Same rationale applies to Fig. 3C, and Fig. 3D-E. In other words, and if I understood correctly, the authors show that the proportion chromatin:soluble is 80:20 after MG132 treatment; however, if this proportion would be the same *before* MG132 treatment, the conclusion would be that the ubiquitylation is happening as efficiently in chromatin as in the soluble fraction, correct? Thus the importance to compare the results presented with results *before/without* MG132 treatment.
- The subtitle "RNF12 chromatin recruitment is required for patterning gene expression" seems hard to justify when the authors look at the expression of only one gene (Xist) and not when its expression is most significant and dynamic (during differentiation of XX mESC). Besides, what do the authors mean with "patterning"? It appears at several places in the manuscript but not sure it is justified.
- The sentence "We show that RNF12 is recruited to chromatin in a spatially restricted manner" is maybe misleading - is the restriction related to space or related to sequence? Such expression also appears at the end of the Introduction.
- In the Discussion the authors state that "We show that recruitment to these specific locations facilitates RNF12 mediated ubiquitylation of REX1, thereby patterning expression of RNF12-REX1 dependent genes, including the key long-noncoding RNA Xist and its antisense transcript Tsix, which collectively coordinate X-chromosome inactivation." The sentence should be rephrased, since it can suggest (and therefore be misleading) that the authors checked expression of Tsix and other RNF12-REX1-dependent genes besides Xist.
- When the authors say, in the Discussion, that "the RNF12 BR appears to make distinct contacts with chromatin and REX1, which together facilitate REX1 ubiquitylation on chromatin." - on which evidence is the first part of the sentence based?
- The authors refer to AlphaFold predictions of the RNF12 structure - why is it not shown or cited?
- The references format should be double checked; I noticed that for one of them (#14), it reads "... Hall, L.L., and Green, M.R. (2010)...." when actually there are still four more authors in the paper after Green M.R.

Referee Cross-Comments

The comments made by Reviewer #1 are very relevant and I agree that they should be addressed.

Reviewer #1 (Comments to the Authors (Required)):

The manuscript by Daudén et al focuses on deciphering how the E3 ligase activity of RNF12, a RING E3 ligase, is regulated. Initial experiments focus on identifying RNF12 partner proteins. The initial data obtained leads them to focus on the nuclear fraction as a key substrate of RNF12 is Rex1, which is a transcription factor with an important role in development. Although they take an untargeted approach their focus is on the known substrate protein Rex1, which the Findlay group have analysed previously (e.g. Bustos et al 2018). Using various assays they identify details of chromatin recruitment of RNF12, including a preferred binding site.

Following establishment of a cellular assay that enables them to assess Rex ubiquitylation in cells the authors utilise a number of deletion constructs to probe to function of different parts of RNF12. Based on these studies they conclude:

- The N-terminal 200 residues inhibit chromatin recruitment
- The basic residues are important for chromatin recruitment & Rex1 ubiquitination
- These two regions therefore have opposing effects on chromatin recruitment, substrate binding and ubiquitination.

The manuscript is relatively well written and will be of interest to those interested in the function of RNF12. However, my major concern is the reliance on deletion studies. While additional studies would be ideal, the reader should be made aware of this potentially significant caveat.

We thank the referee for their comments, and provide new data in which we analyse the function of the RNF12 Basic Region in isolation, to provide orthogonal validation of the deletion studies (see details below).

Main points:

1) In figure 2 the authors use the RNF12 H569A/C572A mutant as a control. This mutant will result in unfolding of the RING domain which will likely influence partner protein binding. The RNF12-W576Y mutant may have been a better control. This should be commented on.

We agree that RNF12 W576Y is a better control, as RNF12 H579A/C572A likely leads to RING unfolding. We have commented on this in the text (p7 paragraph 1). We also provide new data (Fig. 3E,F) that the RNF12 H579A/C572A and W576Y catalytic mutants behave similarly with respect to REX1 ubiquitylation and chromatin recruitment (p9 paragraph 1).

2) Deletion studies are used throughout the manuscript to attribute functions to different parts of RNF12. While these studies provide clues about function, significant caveats remain as deletion studies of this type end up bringing distant sequences into close proximity (i.e. adjacent!) and this can create novel functions. This is a significant concern and it would be helpful if the deletion studies were supported by other data i.e. point mutations or assessment of the proposed domains i.e. does the Basic domain alone bind to chromatin?

We agree that there are caveats associated with use of deletion mutants. We now provide new data (Fig. 4D) showing that isolated RNF12 basic region is efficiently recruited to chromatin, whilst the isolated N-terminal region is not, despite being present in the nucleus (Fig. S2E). This confirms that the RNF12 basic region is necessary and sufficient for RNF12 chromatin recruitment, and supports the results obtained from deletion studies.

3) In a number of the experiments/figures the authors rely on fractionation of proteins into nuclear and soluble fractions. It would be helpful to show in the supplementary material the total protein expression as well as the protein in the nuclear/soluble fractions. This would help give the author a better idea of protein levels and ascertain if any of the deletion proteins cause protein aggregation/mislocalised.

We provide new data analysing RNF12 deletion mutant expression in soluble and chromatin fractions, and in total cell extracts (Fig. S2B, S2D). We also show by immunofluorescence that the relevant deletion constructs are correctly localised (Fig. S2C), confirming our previous observations with a similar series of RNF12 deletion constructs (Segarra-Fas et al, 2022).

4) Quantification of Rex1 ubiquitination in Figure 5d looks challenging. The details of how this was

done should be outlined in the figure legend. For example it would be helpful to know how the background intensity in the -Rex1 samples was treated as the -Rex1 samples for the different RNF12 proteins vary quite a bit. It may be better to have probed Rex1 as done in Fig 3a/6d. This figure needs further explanation.

We agree with the referee that quantification of REX1 ubiquitylation is challenging. We have now modified all relevant figures to highlight the specific REX1 ubiquitylated bands that were quantified and clarified in the figure legend the methods used to measure REX1 ubiquitylation.

In the specific case of Fig. 5D, using the REX1 antibody as in Fig 3A/6D is difficult because the background signal makes it very difficult to identify REX1 ubiquitylated species (see REX1 immunoblot in Fig. 5D). Instead, Cy5 fluorescently-labelled ubiquitin provides a very specific ubiquitin signal, from which the first three REX1 ubiquitylated species (REX1-Ub³) were identified by careful comparison of control and REX1 samples. However, background correction is not appropriate for these assays; although RNF12 preferentially ubiquitylates REX1, in the absence of substrate RNF12 performs auto-ubiquitylation and/or forms free ubiquitin chains, creating variability in the control signal, as pointed out by the referee. We have now modified Fig. 5D to highlight the specific REX1 ubiquitylated bands that were quantified and clarified how REX1 ubiquitylation was quantified in the legend for Fig. 5E.

Minor points:

1) It would be better if panel 4c was below/adjacent to panel 4b.

For consistency we have kept the formatting as there are several other figures organised in this way. We hope that the Figure legends clearly indicate that Fig 4C is quantification of data represented in 4B.

2) Page 21, line 5: In the current format ref 35 should be referred to by first author.

Thank you, we have now made this alteration and checked formatting for other references.

3) It would be helpful to include the sequence of RNF12 - at least the basic region which they state is conserved, but don't show, in the supp material.

We now show a full alignment of RNF12 across vertebrate species and highlight the conserved basic region (Fig. S2A)

4) Figure 4f/g is not intuitive and the authors may wish to consider how to improve clarity.

Thank you to the referee for this suggestion. We now provide a clearer figure showing RNF12 chromatin association in RNF12 WT, KO and RNF12:REX1 DKO mESCs (Fig 3F-G). We have moved the original Fig. 4F/G containing the RNF12 WT and deletion mutants to the supplementary materials (Fig. S2F,G).

5) In the discussion the authors refer to the alphafold model and interactions. It would be good to include this as a supplementary figure.

Many thanks for this suggestion. We now provide an Alphafold structure of RNF12 illustrating predicted interactions between the Basic Region and N-terminal region (Fig. S3).

Reviewer #2 (Comments to the Authors (Required)):

In this study, the authors explore molecular underpinnings of the function of the E3 ubiquitin ligase RNF12/RLIM. TurboID analysis to identify proximal interactors of RNF12 seems to unravel a high proportion of chromatin-associated factors, suggesting that RNF12's functions are associated with chromatin, which the authors went on to explore. ChIP-seq of RNF12 revealed discrete binding across the genome, most but not all overlapping with its main target, the transcription factor REX1. Consistently, the authors claim that REX1 is specifically ubiquitinated on chromatin, which seems to be the case but this part needs to be clarified (see below). The authors also showed that RNF12 associates with chromatin in the absence of REX1. Moreover, the authors functionally dissected different protein domains of RNF12 to identify which ones are responsible for the association to chromatin and/or ubiquitylation functions - the emerging picture is that RNF12 associates with chromatin via a domain rich in basic residues, and the chromatin recruitment is important for ubiquitylation of REX1, while N-terminal sequences have a suppressing effect on chromatin recruitment and subsequent substrate ubiquitylation. Finally, the authors show the importance of the basic residue region for RNF12 function by analysing expression of Xist, an RNF12-REX1 target, in XY mESC using an overexpression assay - while overexpression of wildtype RNF12 leads to Xist upregulation, this is not the case for RNF12 versions without the basic region.

This is an interesting study, but there seems to be overinterpretation of results and the manuscript could also be improved for clarity and accuracy; for instance, as pointed out below, several statements in the Discussion do not seem to be backed up by the evidence shown (or the authors do not make their point come across clearly).

- The title could be improved, the current form is very cryptic, i.e., difficult to understand what it means / what was done and discovered. Moreover, "programmes developmental transcription" sounds too big when the authors checked only one gene (see more comments below). Still in the title and across the paper, the expression "genomic targeting" can be misleading - my first thought was that the authors meant they used a knockout approach, when I suppose they mean chromatin association/recruitment? The strongest point of the paper is the "novelty" of the RNF12 E3 ubiquitin ligase associating with chromatin and I think the title could reflect just that - as the authors mention in the discussion, "we uncover a critical role for chromatin in regulation of substrate ubiquitylation"

Many thanks to the referee for these suggestions. We have now modified the title to 'Chromatin targeting of the RNF12/RLIM E3 ubiquitin ligase controls transcriptional responses'.

- Please explain what MG132 is the first time it appears, and the rationale to use it - for the non-specialists it is not obvious.

Thank you, we have now explained that MG132 is a proteasomal inhibitor that is used to stabilise proteins that might otherwise be degraded (p4 paragraph 2).

- In the introduction, the authors state that "RNF12 regulates highly specific gene expression programmes involved in X-chromosome inactivation, neurodevelopment and gametogenesis" - what about the functions in the mammary gland? (at least one example in PMID: 22541433)

We apologise to the referee for the omission of this important RNF12 function from the introduction. We have now explained that RNF12 regulates mammary gland development and function and added the reference (p2 paragraph 2).

- In Fig. 1B, the nuclear localisation of HA-TurboID-RNF12 is compared to HA-TurboID but should be compared to RNF12?

We show that HA-TurboID RNF12 is localised to the nucleus (Fig. 1B), which reflects RNF12 nuclear localisation reported in most contexts. We also provide new data comparing RNF12 staining in HA-TurboID and HA-TurboID-RNF12 (New Fig. 1B, previous Fig. 1B removed).

- The authors show that there is little overlap between their TurboID approach and the RNF12 affinity-purification mass-spec approach done previously by Gontan et al 2012, and state that this suggests that "the TurboID approach is able to reveal previously undiscovered RNF12 proximal

proteins". Could it suggest instead that the approach picks up a lot of "noise"? How can these two possibilities be distinguished?

To address this point, we have obtained orthogonal evidence that RNF12 interacting proteins are enriched for chromatin-associated factors. New analysis of RNF12 interacting proteins previously identified by affinity purification mass spectrometry (AP-MS; Table S4) indicates that this cohort is also enriched for chromatin associated factors (Table S5). Differences in the individual proteins identified are likely explained by comparing different approaches and biological context i.e. direct interactions (AP-MS) versus proximity (TurboID) and differentiating female ES cells with high RNF12 expression (AP-MS) versus pluripotent male ES cells (TurboID). We have modified the text to incorporate this data and accompanying explanation (p5 paragraph 2).

- Related to the previous point, the authors consider results of the TurboID approach only based on fold-change (higher than 2) and dismissing statistical significance. What is the rationale for this? Shouldn't the authors only consider hits for which the p-value is below at least 0.05? How does this affect the conclusion, i.e., that there is a high proportion of factors associated to chromatin?

We thank the referee for raising this important point. We have now listed all statistically significant RNF12 proximal proteins identified by TurboID ($p < 0.05$) (Table S3). Although the dataset is too small to determine significantly enriched functions, several known chromatin-associated factors are identified (highlighted). As discussed above, we have also obtained orthogonal evidence for RNF12 chromatin association, namely that analysis of RNF12 AP-MS data indicates that RNF12 interacting proteins are enriched for chromatin-associated factors (Table S5).

- Could the authors add a line explaining the choice of DAVID (instead of GO or GSEA, for instance)?

Thanks to the referee for this suggestion. We now explain that we employed DAVID analysis, which incorporates gene ontology (GO) terms and is more practical for analysis of focussed datasets such as this one, in comparison to gene set enrichment analysis (GSEA) (p5 paragraph 2).

- Still regarding Fig. 1D and the DAVID gene set enrichment analysis: RNF12 proximity labelled proteins are significantly enriched for chromatin-dependent functions - compared to...? What is the subset of proteins being compared to? Given that the authors already pre-selected proteins with annotated nuclear localisation and/or function, was the "control group" pre-selected for this as well? Otherwise, it is very likely that a group of proteins selected for nuclear localisation and/or function will be enriched in chromatin-related functions when compared to all proteins?

As discussed above, we address this point by analysis of RNF12 AP-MS data, which indicates enrichment of chromatin-associated proteins without pre-selection of nuclear proteins (Table S5).

- Could the authors comment on the fact that more than half of the sites bound by RNF12-WT and RNF12-H569A,C572A do not coincide? Isn't this unexpected?

We thank the referee for pointing this out and agree that this unexpected finding warrants further investigation. We provide new quantitative analysis of ChIP-seq data that goes beyond peak threshold assignment to investigate the extent of differences between RNF12 WT and RNF12 H569A/C572A chromatin recruitment (New Fig. 2D; previous Fig. 2D moved to Fig. S1A). This analysis shows that most peaks are common between REX1, RNF12 WT and RNF12 H569A/C572A, but also reveals REX1 unique peaks and RNF12 unique peaks (New Fig. 2D, Fig. S1B-D) as suggested by our initial analysis. Importantly, we find that WT and RNF12 H569A/C572A largely occupy the same genomic regions. We propose that the initial difference observed between RNF12 WT and RNF12 H569A/C572A may have resulted from the different relative stabilities of the RNF12 WT and RNF12 H569A/C572A.

- Given the central message of the paper being RNF12 association to chromatin, I think it would be worth exploring further the RNF12 "binding" sites. For sites where there is no Rex1, are there other factors enriched? (based on published datasets and/or motif analysis?) How would this compare to other RNF12 targets?

We again thank the referee for this interesting suggestion and provide exciting new analysis of transcription factor motif enrichment at RNF12- or REX1-unique chromatin binding sites (Fig. S1E).

This shows distinct predicted transcription factor binding profiles between the two sets of genomic sites, suggesting that different sets of transcription factors may be involved in regulation of genes that are bound by RNF12 alone.

- The authors report a "strong enrichment of RNF12 and REX1 at transcriptional start sites" - are these the same TSS for both RNF12 and REX1? What is the overlap?

We address this point in Fig. 2D, in which each row represents the same gene across three experimental conditions (REX1, RNF12 WT and RNF12 H569A/C572A). This data suggests a strong overlap between RNF12 and REX1 recruitment at the same transcriptional start sites.

- Could the authors add in the Introduction and/or Discussion what is known - or not known - about how RNF12 and Rex1 physically interact, if at all?

We have now stated in the introduction that RNF12 and REX1 directly interact via sequences in the RNF12 N- and C-terminal regions in the introduction (p3 paragraph 1).

- The Results section about Fig. 3 is difficult to understand for lack of explanations and details. The authors say they measured "ubiquitylation of REX1" (both in the Results sections and Figure legends) but how was this done? Did they use an antibody specific for ubiquitylated REX1? I don't think this exists? And there is no mention of such in the Methods. So I assume that in Fig. 3B the authors are simply quantifying REX1 signal (and not ubiquitylated REX1), except that it's in the presence of MG132?

We agree with the referee that our methods to quantify REX1 ubiquitylation require further explanation. We have now modified all relevant figures to highlight the specific REX1 ubiquitylated bands that were quantified and clarified in the figure legend the methods used to measure REX1 ubiquitylation.

The referee is correct that there is no specific antibody to detect REX1 ubiquitylation. To quantify REX1 ubiquitylation on chromatin, we treat cells with MG132 to prevent degradation of ubiquitylated REX1 and immunoblot with a total REX1 antibody. We then specifically quantify REX1 ubiquitylated species, which are distinguished as a series of distinct bands migrating at higher molecular weight than unmodified REX1, and not unmodified REX1. We have clarified this point in the text (p9 paragraph 1) and figure legend (Fig. 3B).

- Moreover, for a proper assessment of amount of ubiquitylated REX1 in the chromatin fraction compared to soluble, and to conclude that there is a chromatin-specific effect, I think the results from Fig. 3A have to be compared to results without MG132 treatment (maybe results from Fig. 2C could be used?)? Same rationale applies to Fig. 3C, and Fig. 3D-E. In other words, and if I understood correctly, the authors show that the proportion chromatin:soluble is 80:20 after MG132 treatment; however, if this proportion would be the same *before* MG132 treatment, the conclusion would be that the ubiquitylation is happening as efficiently in chromatin as in the soluble fraction, correct? Thus the importance to compare the results presented with results *before/without* MG132 treatment.

In the absence of MG132, ubiquitylated REX1 is almost undetectable (Fig. 3A), presumably because it is degraded by the proteasome. However, we provide new data showing that MG132 treatment not only increases the absolute amount of ubiquitylated REX1 found on chromatin, it also increases the proportion of ubiquitylated REX1 on chromatin compared to soluble (Fig. 3B).

- The subtitle "RNF12 chromatin recruitment is required for patterning gene expression" seems hard to justify when the authors look at the expression of only one gene (*Xist*) and not when its expression is most significant and dynamic (during differentiation of XX mESC). Besides, what do the authors mean with "patterning"? It appears at several places in the manuscript but not sure it is justified.

We thank the referee for this suggestion. We have now tempered our conclusions to state that RNF12 chromatin recruitment is required for transcription of the X-chromosome inactivation factor *Xist* (p15 paragraph 2). We also remove the term 'patterning' from the manuscript (e.g. p3, p13, p14, Fig. 7 legend)

- The sentence "We show that RNF12 is recruited to chromatin in a spatially restricted manner" is maybe misleading - is the restriction related to space or related to sequence? Such expression also appears at the end of the Introduction.

We thank the referee for this suggestion, and have modified this terminology to state that RNF12 is recruited to specific DNA sequences (e.g. p3 paragraph 2 & p15 paragraph 1).

- In the Discussion the authors state that "We show that recruitment to these specific locations facilitates RNF12 mediated ubiquitylation of REX1, thereby patterning expression of RNF12-REX1 dependent genes, including the key long-noncoding RNA Xist and its antisense transcript Tsix, which collectively coordinate X-chromosome inactivation." The sentence should be rephrased, since it can suggest (and therefore be misleading) that the authors checked expression of Tsix and other RNF12-REX1-dependent genes besides Xist.

We thank the referee for highlighting this confusing sentence. We have now removed the reference to *Tsix* and state that 'We show that recruitment to these specific locations facilitates RNF12 mediated ubiquitylation of REX1, thereby inducing expression of RNF12-REX1 dependent genes, as measured by the key long-non-coding RNA *Xist*, which coordinates X-chromosome inactivation (p16 paragraph 1)

- When the authors say, in the Discussion, that "the RNF12 BR appears to make distinct contacts with chromatin and REX1, which together facilitate REX1 ubiquitylation on chromatin." - on which evidence is the first part of the sentence based?

We thank the referee for pointing out this misleading sentence. We now clarify as follows; 'Indeed, the RNF12 BR is required for interaction with both chromatin and REX1, which facilitates REX1 ubiquitylation on chromatin' (p16 paragraph 1).

- The authors refer to AlphaFold predictions of the RNF12 structure - why is it not shown or cited?

Many thanks for this suggestion. We now provide an AlphaFold prediction of RNF12 illustrating interactions between the Basic Region and N-terminal region (Fig. S3).

- The references format should be double checked; I noticed that for one of them (#14), it reads "... Hall, L.L., and Green, M.R. (2010)...." when actually there are still four more authors in the paper after Green M.R.

We thank the referee for highlighting this error. We have now corrected the formatting of the references.

Referee Cross-Comments

The comments made by Reviewer #1 are very relevant and I agree that they should be addressed.

We agree and have addressed the comments made by Reviewer 1 as detailed in this rebuttal.

December 18, 2023

RE: Life Science Alliance Manuscript #LSA-2023-02282R

Dr. Greg Findlay
MRC Protein Phosphorylation and Ubiquitylation Unit
School of Life Sciences
Australia

Dear Dr. Findlay,

Thank you for submitting your revised manuscript entitled "Chromatin targeting of the RNF12/RLIM E3 ubiquitin ligase controls transcriptional responses". We would be happy to publish your paper in Life Science Alliance pending final revisions necessary to meet our formatting guidelines.

Along with points mentioned below, please tend to the following:
-please upload your main and supplementary figures as single files
-please label EXPERIMENTAL PROCEDURES as Materials & Methods
-please add a callout for Figure 4E to your main manuscript text

A. FINAL FILES:

B. MANUSCRIPT ORGANIZATION AND FORMATTING:

****It is Life Science Alliance policy that if requested, original data images must be made available to the editors. Failure to provide original images upon request will result in unavoidable delays in publication. Please ensure that you have access to all original**

data images prior to final submission.**

The license to publish form must be signed before your manuscript can be sent to production. A link to the electronic license to publish form will be available to the corresponding author only. Please take a moment to check your funder requirements.

Sincerely,

Reviewer #1 (Comments to the Authors (Required)):

The authors have addressed my concerns and the manuscript appears to be considerably improved.

Reviewer #2 (Comments to the Authors (Required)):

I would like to thank the authors for their efforts in properly addressing all my comments and those of the other reviewer. I hope to see this study published soon. Congratulations!

December 22, 2023

RE: Life Science Alliance Manuscript #LSA-2023-02282RR

Dr. Greg Findlay
MRC Protein Phosphorylation and Ubiquitylation Unit
School of Life Sciences
United Kingdom

Dear Dr. Findlay,

Thank you for submitting your Research Article entitled "Chromatin targeting of the RNF12/RLIM E3 ubiquitin ligase controls transcriptional responses". It is a pleasure to let you know that your manuscript is now accepted for publication in Life Science Alliance. Congratulations on this interesting work.

DISTRIBUTION OF MATERIALS:

Again, congratulations on a very nice paper. I hope you found the review process to be constructive and are pleased with how the manuscript was handled editorially. We look forward to future exciting submissions from your lab.

Sincerely,
